# diaTracer enables spectrum-centric analysis of diaPASEF proteomics data

Kai Li [1], Guo Ci Teo [2], Kevin L. Yang [1], Fengchao Yu [2] ✉ &
Alexey I. Nesvizhskii [1,2] ✉

Data-independent acquisition has become a widely used strategy for peptide and protein quantification in liquid chromatography-tandem mass spectrometry-based proteomics studies. The integration of ion mobility separation into data-independent acquisition analysis, such as the diaPASEF technology available on Bruker's timsTOF platform, further improves the quantification accuracy and protein depth achievable using data-independent acquisition. We introduce diaTracer, a spectrum-centric computational tool optimized for diaPASEF data. diaTracer performs three-dimensional (mass to charge ratio, retention time, ion mobility) peak tracing and feature detection to generate precursor-resolved "pseudo-tandem mass spectra", facilitating direct ("spectral-library free") peptide identification and quantification from diaPASEF data. diaTracer is available as a stand-alone tool and is fully integrated into the widely used FragPipe computational platform. We demonstrate the performance of diaTracer and FragPipe using diaPASEF data from triple-negative breast cancer, cerebrospinal fluid, and plasma samples, data from phosphoproteomics and human leukocyte antigens immunopeptidomics experiments, and low-input data from a spatial proteomics study. We also show that diaTracer enables unrestricted identification of post-translational modifications from diaPASEF data using open/mass-offset searches.

Liquid chromatography-tandem mass spectrometry (LC-MS/MS) is a commonly used method to analyze proteomics data[1]. There are two primary approaches for acquiring high-throughput data: data-dependent acquisition (DDA) and data-independent acquisition (DIA). DDA isolates and fragments each peptide ion individually to produce a tandem mass spectrum (MS/MS) that contains information on the peptide sequence. In contrast, DIA isolates and fragments multiple peptide ions within specific mass and retention time ranges, resulting in more complex spectra[2–4]. As mass spectrometers have become faster and more sensitive over the years, the popularity of DIA has increased. Furthermore, the use of ion mobility (IM) separation, in addition to liquid chromatography, to further separate peptides and reduce co-elution[5] has made DIA an even more attractive strategy. The

Bruker timsTOF mass spectrometer platform, which couples trapped ion mobility separation with the parallel accumulation serial fragmentation (PASEF)[6] technique and supports both DDA (ddaPASEF) and DIA (diaPASEF) modes, has proven to offer excellent performance for a wide range of quantitative proteomics applications.

Although ion mobility separation helps resolve co-eluted peptides, it presents new challenges for computational analyses. Most peptide identification and quantification tools were not initially designed to support additional ion mobility dimensions. We have proposed an efficient solution[7,8] for analyzing timsTOF data in DDA mode, but there are still open questions regarding the analysis of diaPASEF data generated by timsTOF. The first critical step in any DIA workflow is the creation of a so-called "spectral library" containing a

[1]Gilbert S. Omenn Department of Computational Medicine and Bioinformatics, University of Michigan, Ann Arbor, MI, USA. [2]Department of Pathology, University of Michigan, Ann Arbor, MI, USA. ✉e-mail: yufe@umich.edu; nesvi@med.umich.edu

list of peptide ions to be quantified in the acquired DIA data. For each target peptide ion, the library contains its separation coordinates (LC retention time and ion mobility value) and an empirical or predicted fragmentation spectrum (fragment m/z values and intensities) that are used to locate and quantify the peptide ion signal in each of the acquired DIA runs. A spectral library can be built using DDA data acquired from the same or similar samples, often using additional fractionation steps to increase the proteome depth. This remains an effective strategy for building spectral libraries, including for diaPASEF data analysis[9], however, additional efforts are required to acquire DDA data for building the library.

An alternative computational strategy, first proposed by us and implemented in the DIA-Umpire[10] algorithm, is to identify peptides from DIA data directly and then use these identifications to build a spectral library for the subsequent targeted protein quantification step. Known as direct DIA or "library-free" strategy (contrasting with the requirement of having a DDA-based library in some DIA analysis tools[11,12]), it has now become the main mode of analysis of DIA data in a typical shotgun proteomics experiment. Direct identification of peptides from DIA data can be performed using either the spectrum-centric approach of DIA-Umpire[10,13] (also adopted by Spectronaut[14]), which starts with the detection of extracted ion chromatogram (XIC) features to generate DDA-like pseudo-MS/MS spectra[10] that are compatible with conventional search engines such as MSFragger. Alternatively, direct identification can be performed using a peptide-centric data analysis strategy[15–17], including the library-free mode of DIA-NN[9,18]. Furthermore, MSFragger-DIA[19] implements a hybrid strategy in which full DIA MS/MS scans are searched against a protein sequence database[19,20], followed by targeted precursor and fragment peak tracing and rescoring of the top peptide candidates for each MS/MS scan[19]. Both spectrum-centric and peptide-centric strategies take advantage of deep-leaning-based predictions[21,22] of peptide spectra and separation coordinates (LC and IM), but with significant differences in the use of prediction models. In the fully spectrum-centric approach and in the hybrid MSFragger-DIA strategy, predictions of peptide properties are performed only for a restricted set of top scoring peptides identified from the data[19,21], whereas peptide-centric methods such as DIA-NN require *in-silico* predictions for all possible candidate peptides given the user-specific protein sequence database and search parameters. One important advantage of the spectrum-centric DIA-Umpire strategy of deconvoluting full DIA MS/MS spectra into pseudo-MS/MS spectra is that it facilitates analyses requiring a large search space. These include studies focusing on post-translational modifications (PTMs), studies requiring nonspecific (e.g., human leukocyte antigens, HLA or endogenous peptidomics) or semi-enzymatic (e.g., N-terminomic) searches, or when spectra and/or retention times of chemically labeled peptides cannot be accurately predicted (e.g., in chemoproteomics). However, although successful and frequently used for the analysis of DIA data from the Thermo Fisher Scientific Orbitrap and Sciex tripleTOF mass spectrometry platforms, neither DIA-Umpire nor MSFragger-DIA are capable of processing diaPASEF data because of the challenges of detecting XIC features with the additional ion mobility dimension. Although Spectronaut supports direct identification from diaPASEF data the details of the algorithm, to the best of our knowledge, have not been publicly disclosed.

To address the need for an efficient and versatile method for analyzing diaPASEF data, we developed a spectrum-centric tool called diaTracer. diaTracer has been designed to perform highly efficient processing of three-dimensional (m/z, retention time, ion mobility) information encoded in diaPASEF data to detect precursor and fragment ion features, followed by the generation of pseudo-MS/MS spectra that can be easily analyzed with peptide identification tools such as MSFragger[23]. diaTracer was fully integrated into the widely used FragPipe computational platform. We demonstrate the performance of diaTracer using diaPASEF data from triple-negative breast cancer (TNBC), cerebrospinal fluid (CSF), and plasma proteome datasets, using data from phosphoproteomics and immunopeptidomics experiments, and low-input data from a spatial proteomics study. We also show that our strategy enables unrestricted PTM searches via the application of the open/mass-offset mode of MSFragger to diaTracer-extracted files. Overall, our study demonstrates that diaTracer and FragPipe computational platform enable fast and sensitive analysis of diaPASEF data from a wide range of biological applications.

## Results

### diaTracer workflow in FragPipe

diaTracer takes diaPASEF data (as input .d files) and generates pseudo-MS/MS spectra[10] (in mzML format) for DDA-like database searching. It employs a three-dimensional feature detection method that leverages retention time and ion mobility dimensions to enhance signals and reduce noise interference. In our previous study[24], we demonstrated that summing RT frames with closed ion mobilities contributed to the accurate identification of the correct XIC features. Therefore, diaTracer starts with the aggregation of neighboring RT frames, followed by a two-dimensional (m/z-1/K0) adaptive Gaussian feature detection algorithm for the merged frames. The same approach is used to detect all precursor and fragment XIC features. Ion mobility and m/z values of the signals are determined using the apex. An isotope grouping method filters precursor signals and identifies potential charge states in the MS1 data. diaTracer then calculates the Pearson correlation coefficient of the XIC features between all detected precursors and fragments within user-defined ion mobility and retention time tolerance to assemble pseudo-MS/MS spectra (see Methods).

diaTracer was fully integrated into FragPipe for streamlined, direct analysis of diaPASEF data (Fig. 1). FragPipe (https://fragpipe.nesvilab.org) is a comprehensive computational platform that automates all the data analysis steps. The pseudo-MS/MS spectra extracted by diaTracer are searched using MSFragger[25] in the DDA mode, enabling tryptic, semi-tryptic, nonspecific, mass-offset, or even open database searches for comprehensive PTM characterization[26–28]. Peptide-spectrum matching (PSM) with MSFragger[23,26] is followed by deep learning-based rescoring of PSMs using MSBooster[21] and Percolator[29] (default option; alternatively, using PeptideProphet[30]), protein inference with ProteinProphet[31], and FDR filtering (by default 1% FDR at the PSM, ion, peptide, and protein levels) using Philosopher[32]. PTMProphet[33] is enabled for PTM workflows that require site localization (e.g., phosphoproteomics). The spectral library is generated using EasyPQP, followed by the extraction of quantification from the DIA data using DIA-NN[9,18] (alternatively, using Skyline[34]). Furthermore, a hybrid spectral library[10,19] can also be generated from both diaPASEF data and additional ddaPASEF data when available. Users have a choice to run FragPipe using an easy-to-use graphical user interface (GUI) or a command line version of the pipeline that is fully compatible with high-performance computing or cloud-based environments. A data visualization module, FragPipe-PDV (an extension of the PDV viewer[35]), is integrated into FragPipe and provides a convenient way to view pseudo-MS/MS spectra and PSMs. Quantitative matrices (with quantification extracted by DIA-NN) can be conveniently analyzed using FragPipe-Analyst[36] or MSstats[37].

### Performance evaluation: deep proteome profiling

We first sought to assess the performance of FragPipe with diaTracer on a dataset representative of typical, deep proteome profiling experiments. We downloaded a triple-negative breast cancer (TNBC) dataset[38] generated on a timsTOF Pro platform and containing (1) ddaPASEF data from 12 hydrophilic interaction liquid chromatography, HILIC peptide fractions of a pooled TNBC sample and (2) 16 diaPASEF runs on individual tissue lysate samples. In the original manuscript, these data were analyzed using Spectronaut version 18.5

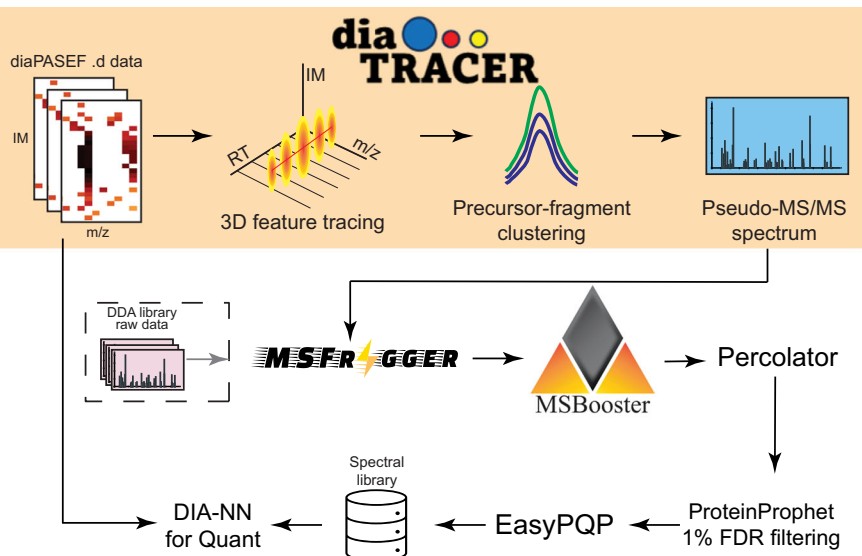

**Fig. 1 | Overview of diaTracer and FragPipe computational platform.** diaTracer applies a 3D feature detection algorithm to detect signals from all possible precursors and fragments in MS1 and MS2 diaPASEF data. Pseudo-MS/MS spectra are generated through precursor-fragment clustering and can be processed as DDA data using MSFragger and FragPipe to build a spectral library directly from the data. A hybrid spectral library can also be generated if DDA data are available. This spectral library is then used to extract quantification using DIA-NN.

using direct DIA approach, using DDA-based library, and using a hybrid (DDA plus direct DIA) library. In addition, the diaPASEF data was processed using DIA-NN 1.8.1 in the "library-free" (direct DIA) mode. We reanalyzed these data using FragPipe using just the 16 diaPASEF files processed with diaTracer (abbreviated "FragPipe" in Fig. 2) and using the hybrid library build from these 16 diaPASEF and the additional 12 fractionated ddaPASEF runs ("FragPipe hybrid"). The Spectronaut 18.5 results (.sne files) and the DIA-NN 1.8.1 library-free results (report.tsv files) were download from the original study.

The number of quantified proteins using the 16 diaPASEF runs alone (direct DIA), was similar between all the tools. Note that the results from all tools were filtered to global protein, global precursor, and run-specific precursor Q-value of 0.01. Under these settings, FragPipe with diaTracer, direct DIA analysis, quantified an average of 9296 proteins per diaPASEF run (Fig. 2a), with 10,341 proteins in total across all runs (Fig. 2b). In comparison, Spectronaut directDIA analysis quantified fewer proteins, 8997 on average per run (10,032 proteins in total). DIA-NN library-free analysis quantified slightly more proteins on average (9520) and in total (10,628) in these data. The number of proteins quantified increased with the use of the ddaPASEF data. Using the hybrid DDA/DIA library, FragPipe again outperforming Spectronaut both in the total number of proteins quantified (11,029 vs.10,552) and per run (9653 vs 9021). Furthermore, FragPipe with the hybrid DDA/DIA analysis strategy quantified the most proteins with no missing values or with <50% missing values across the individual runs (Fig. 2b).

### Cerebrospinal fluid data

We then assessed the performance of diaTracer and FragPipe using the cerebrospinal fluid (CSF) dataset from Mun et al.[39]. This dataset contained 24 ddaPASEF and 34 diaPASEF runs. We performed quantification of 34 diaPASEF samples using the following three FragPipe workflows: (1) Using 34 diaPASEF runs, direct DIA analysis (abbreviated "FragPipe" in Fig. 3). (2) Using the library built from 24 ddaPASEF fractionated runs (abbreviated "FragPipe DDALib"). (3) Using the hybrid library obtained by combining 24 ddaPASEF fractionated runs and 34 diaPASEF runs ("FragPipe HybridLib"). We also used the standalone DIA-NN 1.8.1 in the library-free mode ("DIA-NN lib-free"). Additionally, we used the diaTracer-extracted files to conduct semi-

tryptic and comprehensive PTM searches (using the mass-offset and open search modes of MSFragger). We did not attempt to perform semi-tryptic searches using the DIA-NN library-free mode.

Using the FragPipe workflow (tryptic search), using the 34 diaPASEF files only, we quantified an average of 955 proteins per file, with 1023 proteins in total (Fig. 3a; Supplementary Fig. 1a). In comparison, DIA-NN library-free analysis quantified fewer proteins per file, with 846 proteins on average, despite reporting a higher total number of proteins (Supplementary Fig. 1a). The number of proteins quantified per file using the DDA-based library was 23% (221 proteins) higher than that obtained from direct analysis of the diaPASEF data. Using both ddaPASEF and diaPASEF data (i.e., building the hybrid library) yielded the best results, with an 18% increase (207 proteins) in the average number of quantified proteins per file compared with using the DDA-based library. This also resulted in the highest total number of proteins (Supplementary Fig. 1a) and precursors (Supplementary Fig. 1b) across the four workflows. Interestingly, the average number of quantified precursors per file in the direct DIA analysis with diaTracer was 14% (1217 precursors) higher than that using the DDA-based library (Fig. 3b).

The semi-tryptic search of these data did not result in a significant change in the average number of quantified proteins per file (Fig. 3a) and a minor increase (e.g., 83 in the case of direct DIA analysis) in the total number of proteins (Supplementary Fig. 1a). However, this resulted in a noticeable increase in the number of quantified precursors (39% on average per file) compared to the tryptic search (e.g., 9771 and 13,997 precursors per file, on average, in tryptic and semi-tryptic searches for the direct diaPASEF analysis workflow, respectively). A deeper investigation into the nature of the additional peptides identified in the semi-tryptic search revealed that many of them were derived from immunoglobulin proteins and secreted proteins (Supplementary Data 1). For 158 proteins, we found semi-tryptic peptides mapping to the region immediately following the signal peptides in the N-terminal portion of the corresponding protein (Supplementary Data 2). Figure 3c shows an example of protein hemopexin (HPX), in which the removal of the signal peptide in the mature form of the protein creates a peptide that does not conform to the trypsin cleavage rules. Multiple additional semi-tryptic peptides were identified for this and other proteins (Fig. 3c). Overall, this confirms the presence

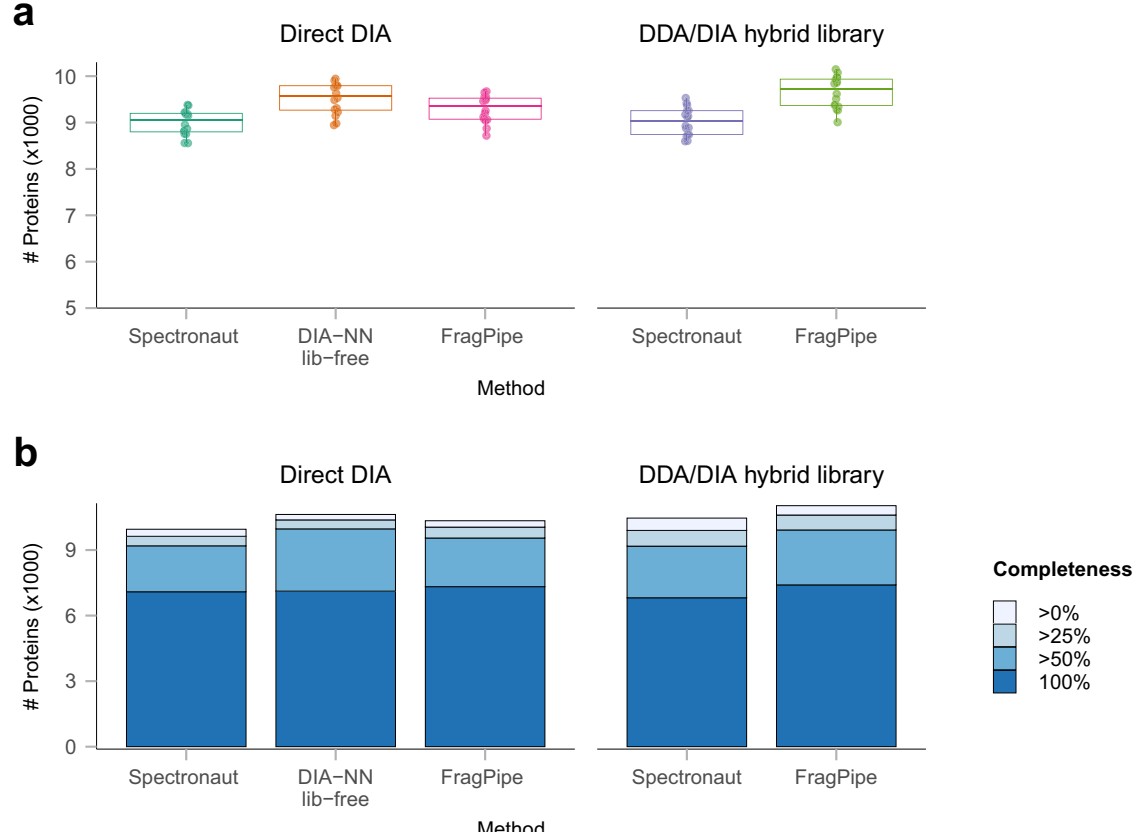

**Fig. 2 | Deep proteome profiling. a** Box plot showing the numbers of quantified proteins of 16 diaPASEF runs from 16 individual TNBC peptide samples using different methods. The lower and upper edges of the box represent the first (Q1) and the third quartiles (Q3). The interquartile range (IQR) is the box between Q1 and Q3. The central line represents the median of the numbers. Whiskers extend from the box to the smallest and largest data points within 1.5 times the IQR from Q1 and Q3, respectively. Data points outside this range are considered outliers and are shown as individual dots. **b** Histogram showing the number of quantified proteins in the TNBC dataset using Spectronaut 18.5 and FragPipe with diaTracer, direct DIA and hybrid DDA/DIA analysis, and using DIA-NN in library-free mode, after application of different non-missing value filters. Shades of blue represent data completeness; darker blues indicate presence in a greater number of samples. Source data are provided as a Source Data file.

of proteolytic cleavage events expected in CSF (and plasma, see below) samples and highlights the importance of monitoring such events using our diaTracer-enabled direct diaPASEF analysis workflow.

We also benchmarked the running time of diaTracer and the rest of the FragPipe workflow (Fig. 3d). diaTracer required 661 min to generate pseudo-MS/MS spectra (19.4 min per diaPASEF file, on average), and the rest of FragPipe (tryptic search) required 87 min to complete the analysis (including running DIA-NN for quantification). In contrast, the DIA-NN library-free analysis took 10.5 min to generate the in-silico predicted spectral library and 1486 min to perform the rest of the analysis, more than twice the time of the diaTracer workflow. Importantly, repeating FragPipe analysis (starting with the existing pseudo-MS/MS mzML files) using the semi-tryptic search took only 110 min. Because diaTracer extracts all possible signals, the generated pseudo-MS/MS mzML file can be reused in any search setting. In contrast, the *in-silico* spectral library generated by DIA-NN can only be reused if the search settings are the same.

We also tested the possibility of performing mass-offset searches using the FragPipe's "Mass-Offset-CommonPTMs" workflow and the same pseudo-MS/MS spectra from diaTracer. Figure 3e displays the most abundant PTMs identified in these data, as summarized by PTM-Shepherd[27]. The list of PTMs included common chemical artifacts, but also biological modifications such as phosphorylation. The analysis took 341 min (or 10 min per file), from MSFragger search to PTM-Shepherd reports. A similar mass shift histogram (Supplementary Fig. 2) was observed from the open search results using the FragPipe's

"Open" workflow, and the computational analysis took similar time (317 min). These mass-offset and open search analyses demonstrate the feasibility of performing comprehensive PTM searches using diaPASEF data processed using diaTracer.

**Plasma proteomics**

Another evaluation was conducted using the plasma dataset from Vitko et al.[40]. In the original study, the authors performed a plasma profiling workflow using Seer's Proteograph Assay Kit, which includes five distinct nanoparticles (NP1-5). We selected 40 diaPASEF runs of NP2 enriched plasma samples analyzed with a 60SPD gradient on a timsTOF HT mass spectrometer. Pseudo-MS/MS spectra were generated from these 40 diaPASEF runs using diaTracer, and the rest of the analysis was completed in FragPipe using MSFragger tryptic and semi-tryptic search settings. The data were also analyzed using the DIA-NN library-free mode. With at least 50% non-missing value filtering (i.e., proteins must be quantified in at least 50% of the runs) and performing tryptic searches, the DIA-NN library-free analysis quantified 2347 proteins, whereas diaTracer coupled with FragPipe quantified 2922 proteins (Fig. 4a). Similarly, although the DIA-NN library-free mode quantified most precursors across all runs in total, the diaTracer workflow in FragPipe resulted in 77% more precursors when requiring a precursor to be quantified in at least 50% of the runs. This demonstrates the improved ability of the diaTracer workflow in FragPipe to consistently quantify peptides across a large sample cohort.

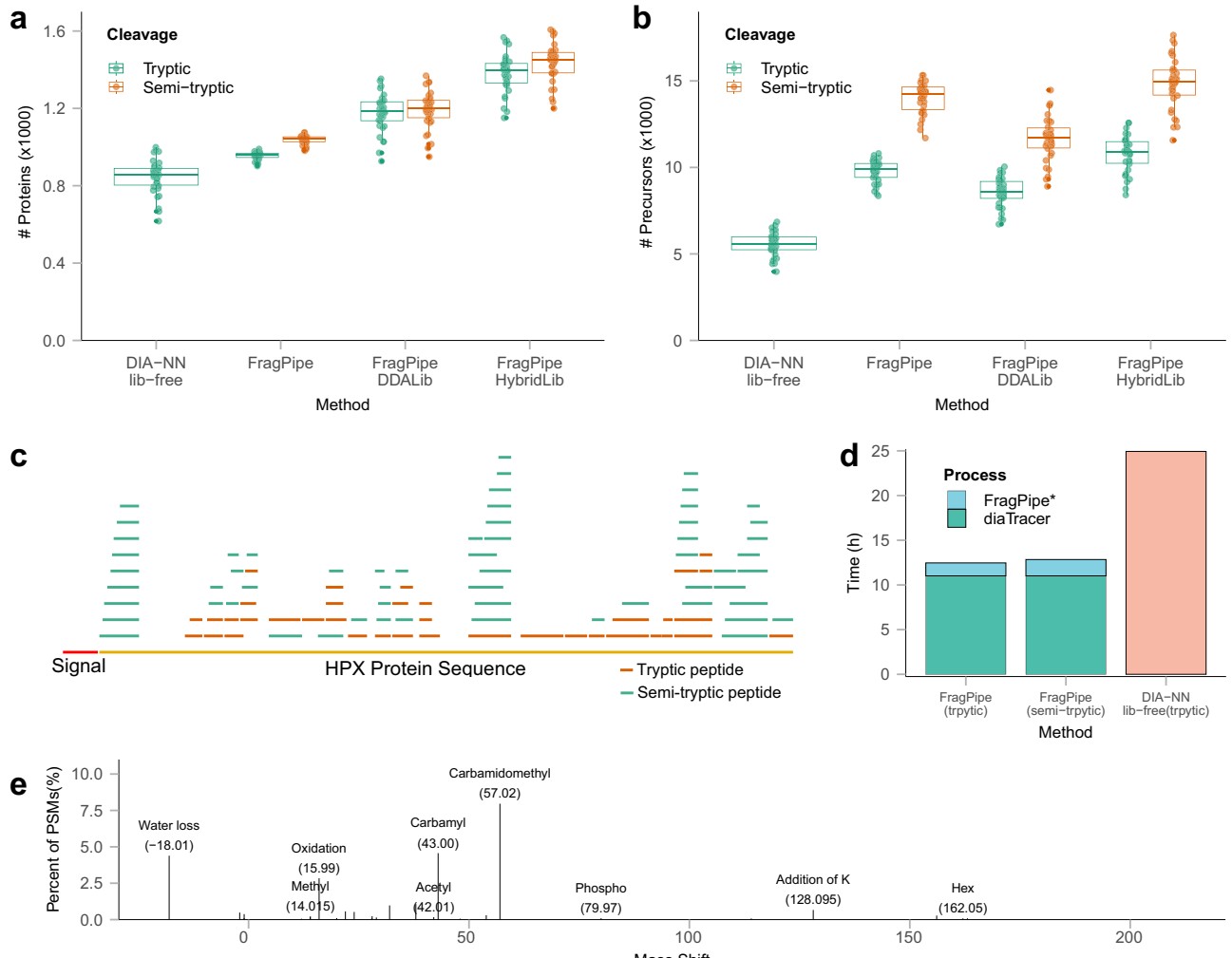

**Fig. 3 | CSF data. a** Box plot showing the numbers of quantified proteins using different methods, with colors representing cleavage types (green: trypsin cleavage; orange: allowing semi-tryptic peptides). Each dot represents the number reported for each of the 34 diaPASEF runs from 15 patients with Alzheimer's disease (AD) and 19 control subjects. In the boxplot, the central line represents the median of the numbers. The lower and upper edges of the box represent the first (Q1) and third quartiles (Q3). The interquartile range (IQR) is the box between Q1 and Q3. Whiskers extend from the box to the smallest and largest data points within 1.5 times the IQR from Q1 and Q3, respectively. Data points outside this range are considered outliers and are shown as individual dots. **b** Box plot showing the numbers of quantified precursors of 34 diaPASEF runs using different methods. The box plots' median, edges, and whiskers are same as the ones in (**a**). **c** Distribution of identified peptides for protein HPX, with the red segment indicating the signal peptide region at the N-terminal. Orange segments represent tryptic peptides, while green segments represent semi-tryptic peptides. **d** Running time comparison. FragPipe* indicated the FragPipe run time (including DIA-NN for quantification) starting from diaTracer-extracted files. **e** Modifications identified using the common mass-offset workflow in FragPipe using pseudo-MS/MS spectra generated by diaTracer. Source data are provided as a Source Data file.

Furthermore, as with the CSF dataset described above, performing a semi-tryptic search significantly increased the number of quantified precursors (but not proteins). Using the semi-tryptic search setting, we identified 30,158 peptides, compared to 26,469 peptides identified using the tryptic search. Of these, 7642 peptides were uniquely identified in the semi-tryptic search (Supplementary Fig. 3). Among these 7642 peptides, 6800 (89%) were semi-tryptic. Additionally, 95% (6458) of the semi-tryptic peptides, corresponding to 1104 proteins, were quantified. Among the 40 samples in this plasma dataset, 20 were from patients with stage IV non-small cell lung cancer (NSCLC) and 20 were from patients in the non-cancer control group. We applied FragPipe-Analyst to conduct differential expression analysis between the NSCLC and control groups using both tryptic and semi-tryptic search results. We detected 69 more significantly differentially expressed proteins from the semi-tryptic results under the same filtering criteria (Supplementary Data 3). In the semi-tryptic results, 35 proteins were upregulated, and 140

proteins were downregulated (Fig. 4c). Figure 4d shows two examples of proteins showing statistically significant upregulation in the NSCLC vs. control group with the semi-tryptic search, for which a non-significant difference was observed with the tryptic search. These two proteins, AKT2 and H2AX, have been previously reported to be associated with NSCLC[41,42]. Upon examining peptide-level quantification, we found the difference in H2AX quantification was due to two semi-tryptic peptides showing significant upregulation in the NSCLC group (Supplementary Fig. 4). For AKT2, however, there were no semi-tryptic peptides quantified. One tryptic peptide identified in the tryptic search was missing from the semi-tryptic search as it did not pass the FDR filters. In the absence of this (low confidence) peptide, AKT2 expression was significantly upregulated in the NSCLC group. Overall, while in-depth interpretation of these examples goes beyond the scope of this work, our results highlight the importance of accounting for proteolytic cleavage events in the analysis of plasma data.

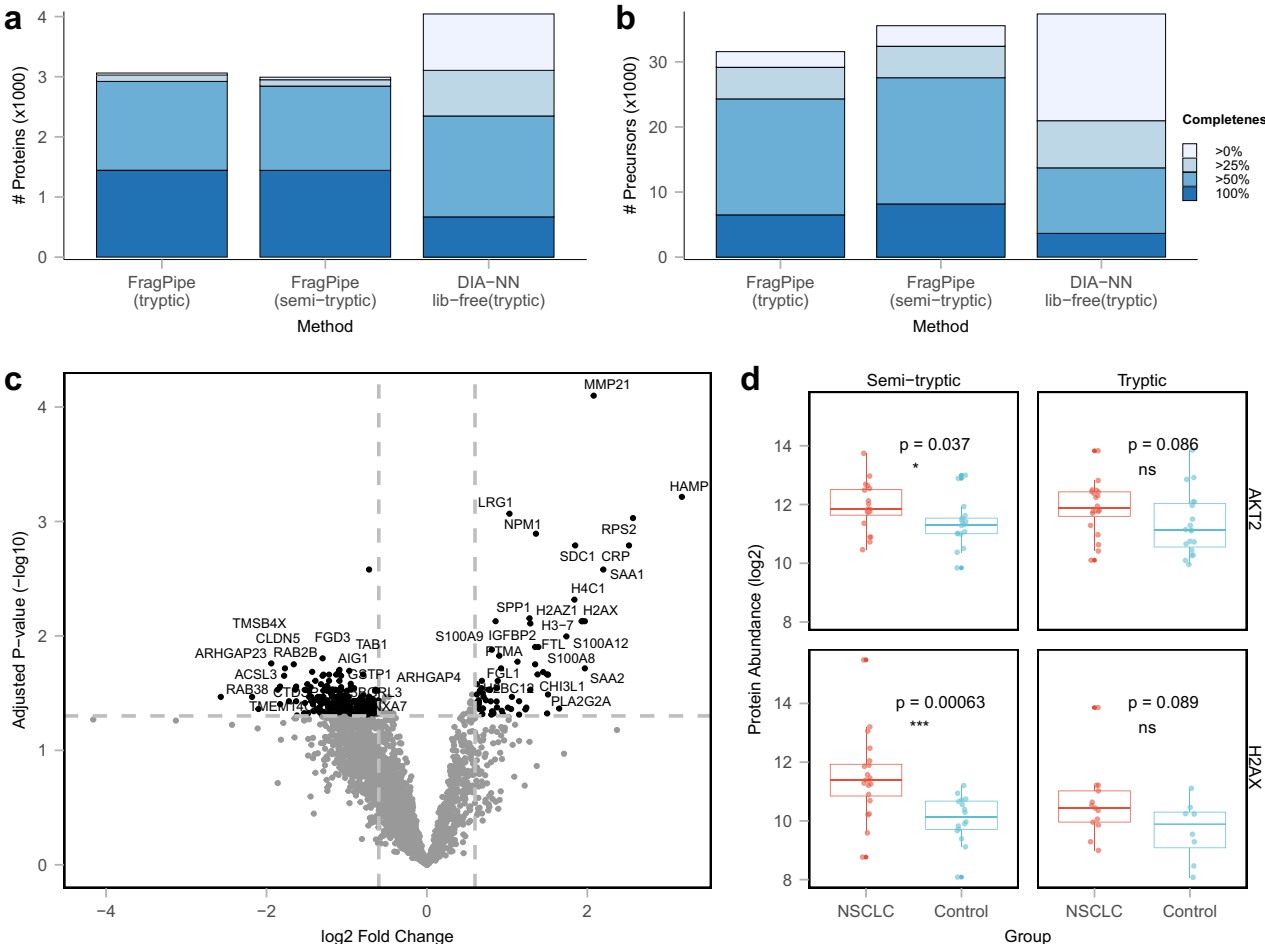

**Fig. 4 | Plasma data. a** Histogram showing the number of quantified proteins using diaTracer-based FragPipe workflows (tryptic and semi-tryptic search), using DIA-NN library-free mode, colored from deep to light blue corresponding to different non-missing value filters. Shades of blue represent data completeness; darker blues indicate presence in a greater number of samples. **b** Same as (**a**) for quantified precursors. **c** Volcano plot comparing protein abundance between stage IV non-small cell lung cancer (NSCLC) and non-cancer control samples, highlighting NSCLC-overexpressed proteins (Log2 fold change ≥ 0.6; adjusted p-value ≤ 0.05). The adjusted p-value is from the moderated t-test followed by the Benjamini-Hochberg procedure. **d** Boxplots of protein abundance distribution of AKT2 (top) and H2AX (bottom) proteins in semi-tryptic (left) and tryptic (right) searches

between 40 diaPASEF runs from 20 NSCLC (red) and 20 control (blue) samples. The p-values change from 0.086 to 0.037 and from 0.089 to 0.00063 for AKT2 and H2AX respectively after using semi-tryptic searches. In the boxplot, the central line represents the median of the numbers. The lower and upper edges of the box represent the first (Q1) and third quartiles (Q3). The interquartile range (IQR) is the box between Q1 and Q3. Whiskers extend from the box to the smallest and largest data points within 1.5 times the IQR from Q1 and Q3, respectively. Data points outside this range are considered outliers and are shown as individual dots. ns: $p > 0.05$; $*p \leq 0.05$; $**p \leq 0.01$; $***p \leq 0.001$; $****p \leq 0.0001$ (two-sided t-test). Source data are provided as a Source Data file.

## HLA immunopeptidomics

Applying peptide-centric strategies to the analysis of HLA or endogenous peptidome data is not practically feasible, given the time required to generate an *in-silico* predicted spectral library for all candidate database peptides under nonspecific protein cleavage rules. The diaTracer approach of extracting pseudo-MS/MS spectra offers a clear advantage for analyzing this type of data. We used an HLA dataset[43] to evaluate the performance of our method and compare it with the results reported in the original study. We selected three replicates of an HLA peptidome sample purified from a healthy donor's plasma sample and analyzed using a Whisper40 gradient (31 min length) on a timsTOF Ultra instrument. As shown in Fig. 5a, from the three replicates, we identified and quantified 2651 immunopeptides with lengths 7–14, which was 18% more than the 2288 peptides reported in the original publication using Spectronaut version 17 directDIA.

To further evaluate our results, we used NetMHCPan-4.1[44] to predict the binding affinity of the quantified immunopeptides with the donor's HLA allele information provided in the original study. We

identified 2327 (90%) strong and 130 (5%) weak binders from 2572 immunopeptides of lengths 8–12 (Fig. 5b). Most of these binders were derived from the HLA-A*24:02 allele (Supplementary Fig. 5), as reported in the original publication. We further summarized the length and charge state distributions of the quantified immunopeptides (Fig. 5c, d). Most immunopeptides had a length of nine and were observed as singly or doubly charged ions. This finding aligns well with the general expectations for HLA data and the specific findings reported for this dataset in the original study. We also compared how the identified immunopeptides overlapped with those reported in the original study using the experimental DDA library and using the pan-library (Supplementary Fig. 6). While it is expected that the analysis of diaPASEF data using a more comprehensive spectral library would result in more peptide identifications, our FragPipe results showed good overlap with the published data. We also selected one peptide to demonstrate the utility of our integrated FragPipe-PDV[35] viewer for inspecting PSMs and MS/MS spectra. The peptide VYQHLFTRI is predicted to be a strong binder. We visualized one of its PSMs along with the predicted spectrum generated by MSBooster[21] (Fig. 5e). The

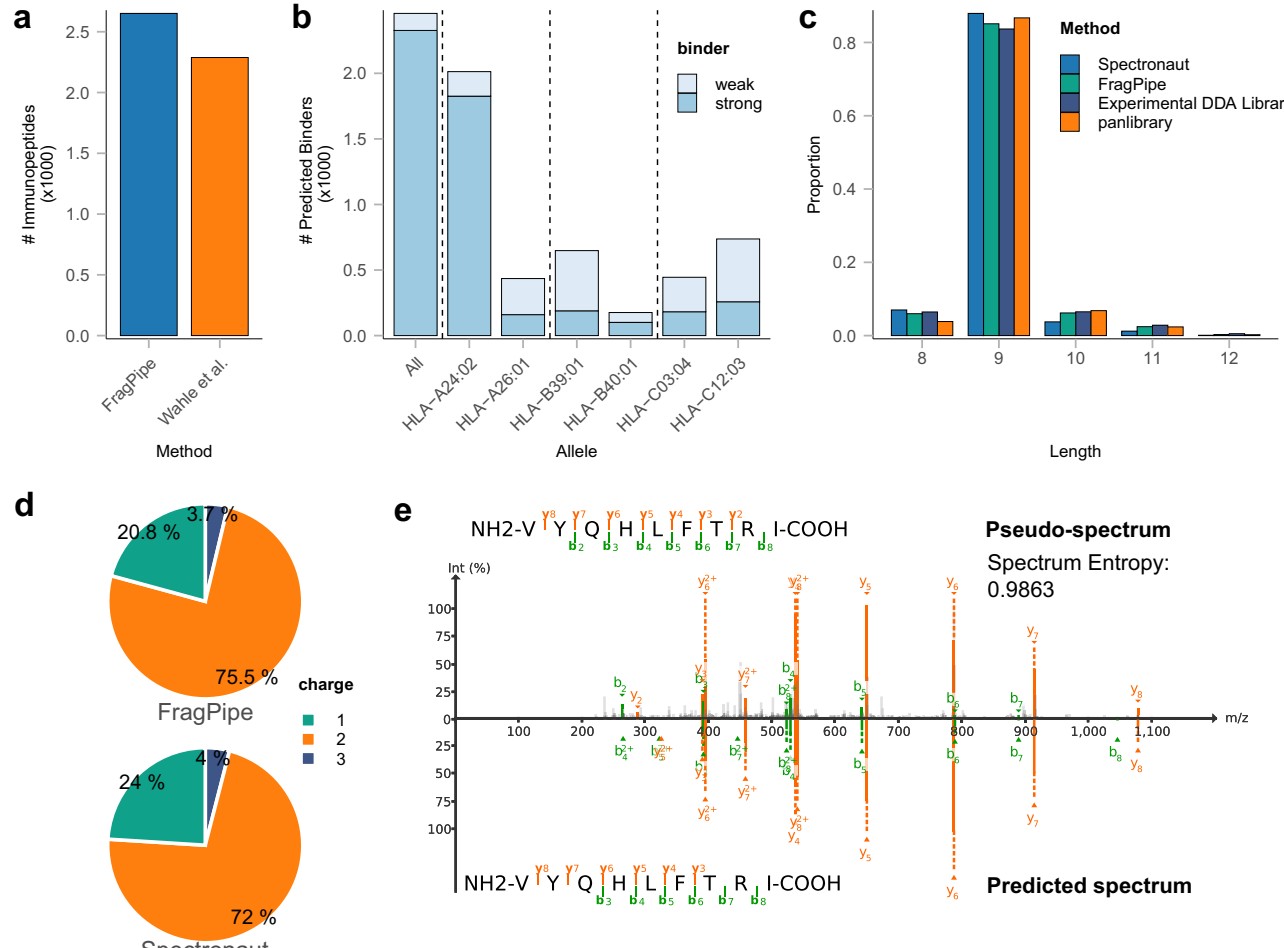

**Fig. 5 | HLA immunopeptidomics data. a** Number of quantified immunopeptides obtained using FragPipe with diaTracer and those reported in the original study based on Spectronaut 17. **b** Histogram of predicted binders for all HLA alleles of the corresponding sample donor, colored by binder type (light: weak binder; dark: strong binder). **c** Length distribution of quantified immunopeptides using FragPipe with diaTracer and that reported in the original study based on Spectronaut 17 directDIA, DDA experimental library, and panlibrary, each with a unique color.

**d** Charge state distribution of quantified immunopeptides using FragPipe with diaTracer and that reported in the original study based on Spectronaut 17 direct-DIA. **e** Pseudo-MS/MS spectrum generated by diaTracer and the predicted spectrum for one of the quantified immunopeptides, VYQHLFTRI. The entropy score between the two spectra is 0.9863. Visualization using FragPipe-PDV viewer. Source data are provided as a Source Data file.

pseudo-MS/MS spectrum extracted by diaTracer exhibited a high degree of similarity (spectrum entropy score of 0.9863) with the predicted spectrum, demonstrating the high quality of the pseudo-MS/MS spectra.

**Phosphoproteomics**

To evaluate the performance of our method in quantifying phosphorylated peptides, we applied our diaTracer-based direct diaPASEF workflow in FragPipe to a phosphorylation-enriched dataset acquired using six gradient lengths[45]. It took diaTracer, on average, 13.5 min to extract signals and produce pseudo-MS/MS spectra for a single 60 min gradient file. For site localization, PTMProphet was enabled, and the site localization information was propagated from the PSM files to the final quantification matrices generated by DIA-NN. We identified 6844 and 11,281 phosphopeptide sequences in the 7 min and 60 min gradient data, respectively (Fig. 6a). In contrast to the original study, where the number of phosphorylated peptide sequences plateaued at a 21 min gradient (Fig. 6a), we observed an increase in the number of quantified phosphorylated peptides with longer gradients. This trend also holds for class I phosphosites (sites with site localization probability >0.75) (Supplementary Fig. 7). We also found that the quantification results reported by diaTracer-based direct diaPASEF workflow

in FragPipe showed high correlations across four technical replicates (Fig. 6b). We also compared the quantification performance with Spectronaut. Supplementary Fig. 8 shows the CV distribution, with box plots illustrating the quartiles and median of the CVs. The result reveal that our tool has comparable quantification precision with Spectronaut in these data.

Although our pipeline computes site localization scores at the spectral library building stage, manual inspection of peptide elution profiles and site localizing fragments observed in a specific sample may be desired. In addition to the FragPipe-PDV viewer mentioned above, the output from FragPipe is fully compatible with the Skyline computational environment for advanced visualization of peptide precursors and fragment ion signals in diaPASEF data. We visualized the PSMs and XICs of two co-eluted isobaric positional isomers, S(pho)PSPPDGSPAATPEIR and SPSPPDGS(pho)PAATPEIR, quantified in a 60 min gradient run. diaTracer successfully separated these two precursor signals along with their corresponding fragment signals into two separate pseudo-MS/MS scans. Both isomers were identified with a site probability >0.75. They eluted ~35 s apart in retention time and were 0.02 1/K0 apart in the ion mobility dimension (Fig. 6c–f). Several fragment peaks that contributed to the localization of the phosphorylation site were clearly visible in the spectra.

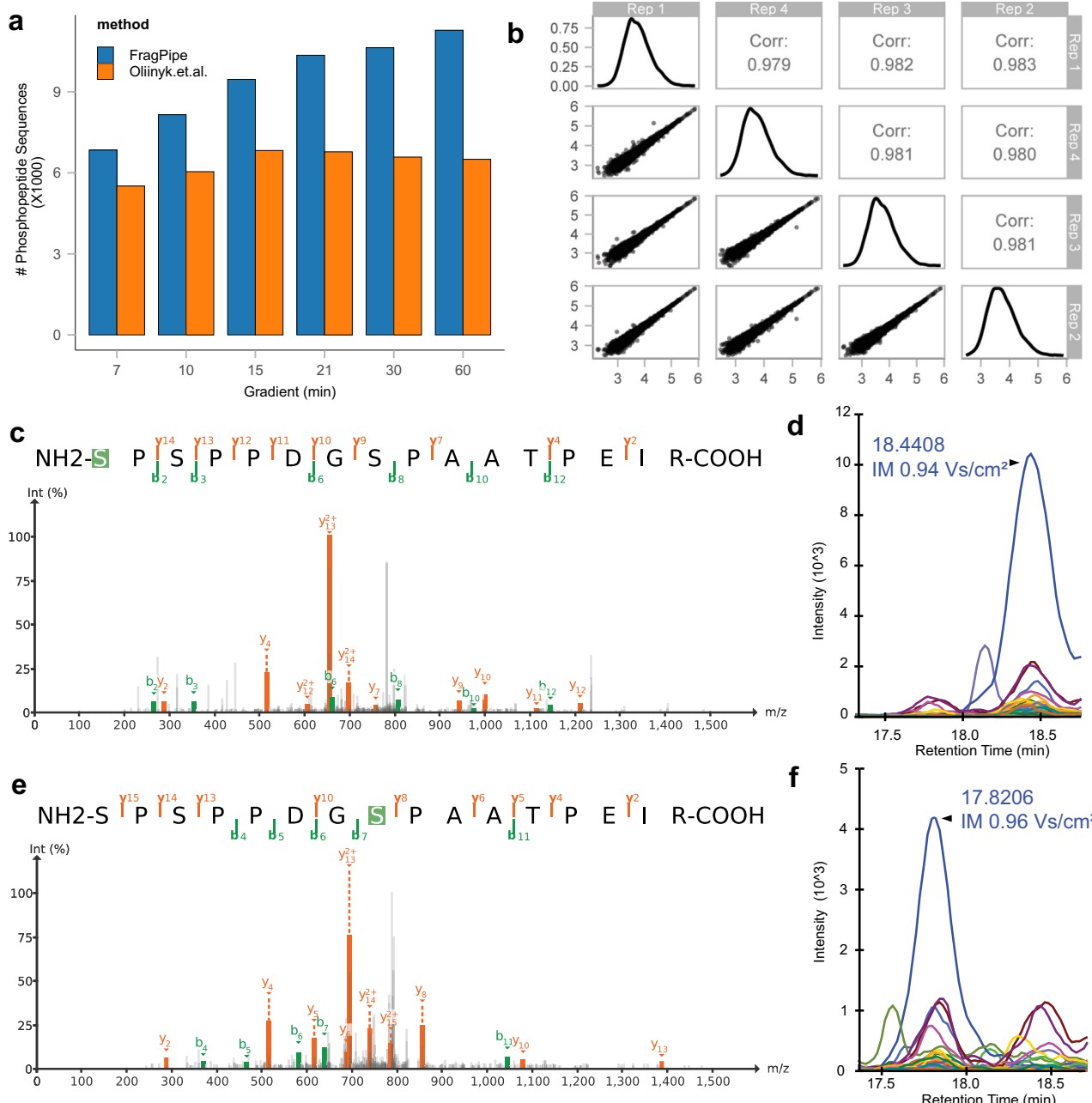

**Fig. 6 | Phosphoproteomics data. a** Histogram of quantified phosphorylated peptide sequences across different gradients using FragPipe with diaTracer (blue) and those reported in the original study based on Spectronaut 16 (orange). **b** Quantified phosphorylated peptides intensities and correlations in four replicates in the 7 min gradient time experiment. **c**, **d** PSM and fragment XICs of phosphorylated peptide S(Pho)PSPPDGSPAATPEIR. **e**, **f** PSM and fragment XICs of phosphorylated peptide SPSPPDGS(Pho)PAATPEIR. The spectrum was generated by FragPipe-PDV. The XICs were generated by Skyline. Source data are provided as a Source Data file.

## Low-input, spatial proteomics data

We then investigated the performance of our method using low-input data from a recent spatial proteomics study[46]. The study profiled 148 microregions from four areas (Epithelium, Germinal center, Mantel zone, and T-cell zone) of a human tonsil obtained using laser capture microdissection (LCM). In the original study, the authors first used the DIA-NN library-free mode to generate a comprehensive human tonsil spectral library using 46 runs acquired in the diaPASEF mode from samples with a high protein amount (referred to as high-input samples below). The authors then applied this high-input spectral library to quantify proteins in the 148 low-input samples. The authors did not attempt to perform a direct analysis of the low-input samples, that is,

without using the high-input spectral library. For comparison with the original study, we performed two analyses using the diaTracer workflow in FragPipe. In the first analysis, we used diaTracer and FragPipe to build the library from the 46 high-input samples ("FragPipe high-input Lib"), similar to the original study that used DIA-NN. Second, we performed a direct analysis of the 148 low-input samples without using high-input data ("FragPipe").

The results of our diaTracer-FragPipe workflow, with high-input data used for building the library, produced identification numbers very similar to those based on DIA-NN library-free analysis, using the reports taken from the original publication (Fig. 7a; Methods). As expected, the high-input library resulted in more proteins, 3349 in

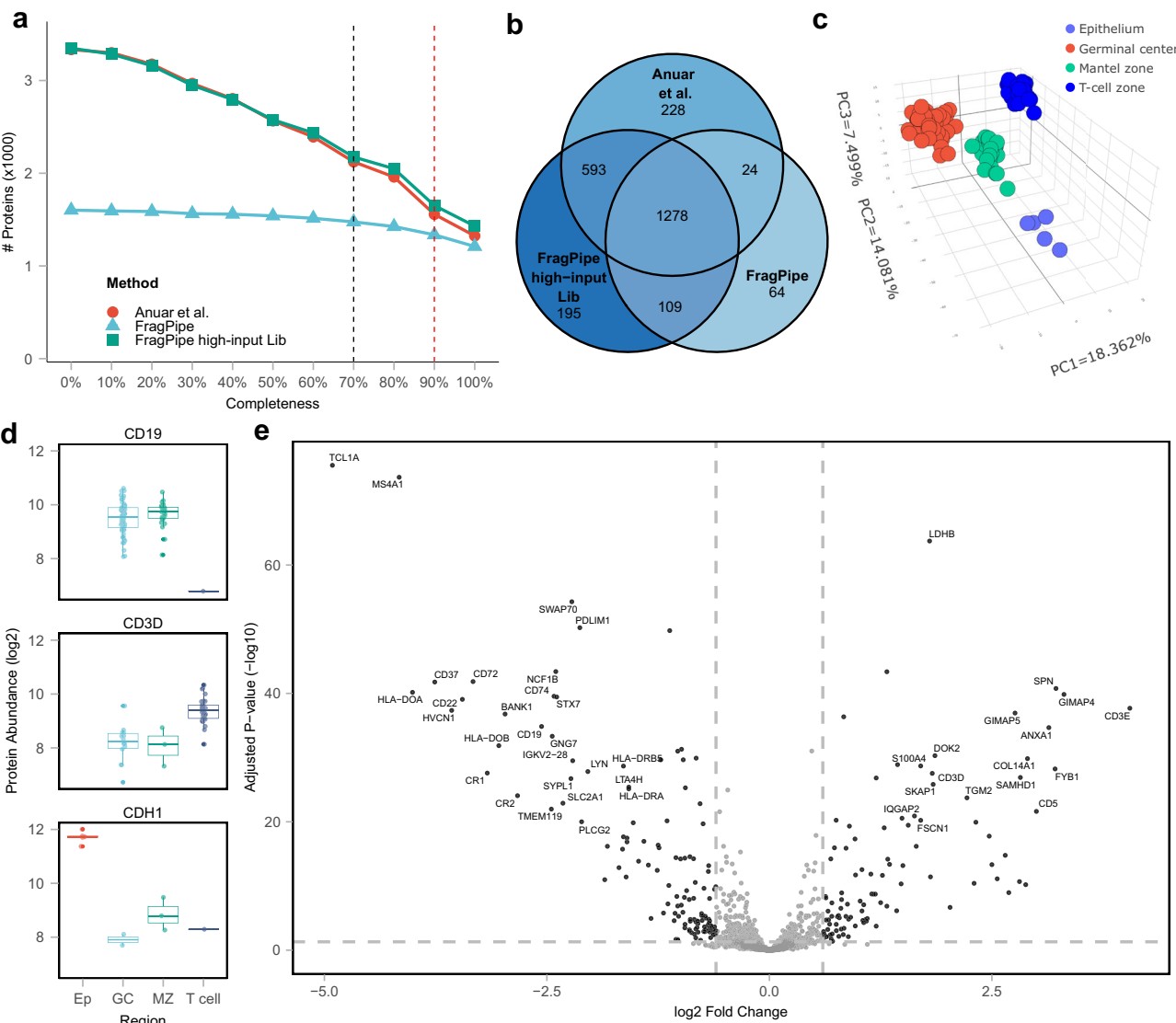

**Fig. 7 | Low-input, spatial proteomics data. a** Number of quantified proteins after application of non-missing value (in at least one group) filter ranging from 0% to 100%, with line colors representing different methods. Red: results from the original study based on the library built using high-input samples; Green: results based on FragPipe with diaTracer, also using high-input data to build the library ("Frag-Pipe high-input Lib"); Blue: result using diaTracer and FragPipe with low-input data only ("FragPipe"). **b** Venn diagram of quantified proteins between the three methods, with data filtered to keep proteins with at least 70% non-missing values in at least one group. **c** Principal-component analysis (PCA) plot of 148 samples based on 1471 proteins (after missing value filtering and data imputation) quantified using the FragPipe workflow (using low-input data only). **d** Log2 transformed protein level abundance distribution of selected cell-type-specific proteins in different regions. In the boxplot, the central line represents the median of the numbers. The lower and upper edges of the box represent the first (Q1) and third quartiles (Q3). The interquartile range (IQR) is the box between Q1 and Q3. Whiskers extend from the box to the smallest and largest data points within 1.5 times the IQR from Q1 and Q3, respectively. Data points outside this range are considered outliers and are shown as individual dots. **e** Volcano plot showing protein abundance differences between Mantel zone (28 samples) and T cell zone (34 samples), highlighting tissue-specific proteins (Log2 fold change ≤ 1; adjusted $p$-value ≤ 0.05). The adjusted $p$-value is from the moderated $t$-test followed by the Benjamini-Hochberg procedure. Source data are provided as a Source Data file.

total (1992 proteins per sample), compared to 1602 proteins (1303 proteins per sample) identified using direct analysis of low-input samples. However, the total number of identified proteins is a misleading metric; proteins detected in one or a few samples have little value for downstream analysis. Indeed, all analyses reported by the authors in the original paper were performed after the application of a missing value filter. More specifically, they required at least 70% non-missing values across all samples in at least one group (the four sample groups mentioned above). With this filter in place, the use of high-input data resulted in 2175 proteins (1692 per sample) compared with 1475 proteins (1265 per sample) quantified directly from low-input samples. Increasing the stringency of the missing value filter further

diminishes the impact of using high-input data. Requiring 90% non-missing values, we identified 1652 (1398 per sample) and 1336 proteins (1182 per sample) with and without using high-input samples alone, respectively.

Figure 7b displays the overlap between proteins quantified with at least 70% non-missing values in at least one class in the original study and using FragPipe with diaTracer with and without high-input sample data. All these approaches exhibited a high degree of overlap. Overall, the biological observations were consistent across all analyses. We specifically investigated whether similar results could be obtained with direct identification from low-input samples, despite quantifying fewer proteins compared to using a larger high-input library. Following the

same data filtering procedure as in the original study, the quantified proteins clearly separated into four distinct cell types (Fig. 7c). We also examined cell type markers reported in the original study, including CD19 (B-cell marker), CD3D (T-cell marker), and CDH1 (epithelial marker), which were all highly expressed in the anticipated sample groups (Fig. 7d). Furthermore, a global comparison between Mantel zone and T cells revealed significant proteomic differences (Fig. 7e). The list of differentially expressed proteins derived from the results with and without using high-input data, when submitted to FragPipe-Analyst, produced a similar list of enriched pathways (Supplementary Fig. 9). Overall, our analysis demonstrates that FragPipe with diaTracer can be effectively used to identify and quantify proteins from a large cohort of low-input diaPASEF samples, even in the absence of high-input data (although having such data is beneficial), especially if focusing on a subset of proteins consistently identified across a large fraction of the samples in the cohort.

## Discussion

We presented a computational method and software, diaTracer, which can deconvolute diaPASEF data to produce DDA-like pseudo-MS/MS spectra. These pseudo-MS/MS spectra can be searched using conventional DDA peptide identification tools, such as MSFragger. diaTracer is available as a stand-alone tool and is integrated into FragPipe, enabling a complete diaPASEF data analysis workflow, including peptide identification, deep-learning-based rescoring, protein inference, FDR filtering, PTM site localization, spectral library generation, quantification, and data visualization. We used several datasets from different types of biological studies to demonstrate the speed and high sensitivity of our computational strategy for analyzing diaPASEF data.

diaTracer enables direct, spectrum-centric analysis of diaPASEF data, which is more versatile than the peptide-centric approach. Advances in deep learning methods have enabled the creation of tools that can predict fragment intensities, ion mobilities, and retention times of peptides with high accuracy, thereby facilitating the generation of in-silico spectral libraries. However, current deep learning prediction tools have limitations. Peptide-centric strategies coupled with in-silico library prediction work well for conventional experiments involving quantitative protein profiling of cell lines and tissues. In contrast, their application to endogenous peptidome or immuno-peptide data is generally not feasible because of the large number of peptide candidates that require predictions. Even semi-tryptic searches, which are often beneficial for certain samples, such as plasma samples and samples from N-terminomics[47,48] studies, remain challenging. Furthermore, peptide-centric analyses struggle with searches involving more than a few common PTMs. Even for phosphorylation, a PTM that receives a great deal of attention, the peptide-centric approach can be computationally prohibitive. Furthermore, for peptides harboring less common PTMs or chemical labels that are not included in the training set used to build prediction models there is an additional risk of poor prediction accuracy.

In contrast, the spectrum-centric approach of DIA-Umpire, and now diaTracer for diaPASEF data, does not have these limitations. After diaTracer generates pseudo-MS/MS spectra, well-established DDA peptide identification tools can be used to perform semi-enzymatic or nonspecific searches. We demonstrated that the semi-enzymatic search provides a significant boost in the number of quantified precursors. It also increases the accuracy and precision of quantification, thereby identifying more differentially expressed proteins in the analysis of CSF and plasma datasets. We also illustrated the high sensitivity and accuracy of our strategy in the analysis of HLA immunopeptidome data. We note that spectrum-centric strategies still benefit from deep-learning predictions and rescoring; however, predictions need to be made only after the search of pseudo-MS/MS spectra and for a relatively small number of peptides. The accuracy of the predictions is also

less critical (e.g., a very inaccurate retention time prediction would not automatically exclude the peptide from being identified), making the pipeline more robust for peptides harboring less common PTMs or chemical labels. Thus, we hope that our strategy will be especially useful for chemoproteomics applications involving different types of chemical labels, such as affinity-based proteome profiling (ABPP) workflows[49,50]. Finally, the spectrum-centric approach enables unrestricted PTM analysis via open or mass-offset searches of pseudo-MS/MS spectra, which is currently not possible using the peptide-centric approach.

The spectrum-centric approach of diaTracer (and DIA-Umpire) also has limitations. First, diaTracer starts with MS1 feature detection, which means it relies on the detection of precursor peptide's signal in MS1 data to group fragments and build a pseudo-MS/MS spectrum. To minimize this limitation, diaTracer aggregates the neighboring RT frames to amplify the MS1 signals. This helps to extract weak signals and filter out noise. Second, diaTracer extracts multiple features for each signal, including ion mobility range, m/z, precursor XIC, XICs of the corresponding fragments and their correlation with the precursor and with each other. However, this information is currently used only at the pseudo-MS/MS generation step. Additionally using it in the subsequent database search and scoring, as it is done in e.g. MSFragger-DIA[19], would likely improve the results. We plan to address these limitations in future work.

The computational time required for processing diaPASEF data is an important consideration. In our tests of diaTracer and FragPipe, the analysis time was less than the time it took to acquire the mass spectrometry data. Furthermore, the conversion of diaPASEF raw files to pseudo-MS/MS spectra can start with the already acquired files while the data acquisition is still ongoing. Finally, the conversion needs to be performed only once. The resulting deconvoluted files can be used in multiple ways, for example, searched using different search parameters, or using different search engines or their combination, all without the need to reprocess the files.

In this study, we focused on the conventional diaPASEF data. At the same time, the rapid technological development of the timsTOF platform led to the introduction of several related data acquisition methods, including Synchro-PASEF[51], Slice-PASEF[52], and midiaPASEF[53]. These techniques aim to increase the efficiency of signal detection by redesigning the schema of the quadrupole selection windows. These new methods better preserve the relationships between the precursor peptide and its fragments, facilitating downstream data analysis. The spectrum-centric approach of diaTracer is well-suited for analyzing data from these new acquisition methods. We plan to extend diaTracer to support these new data types in the future.

## Methods

### diaTracer feature extraction algorithm

The data from one isolation window ($1/k0 \times m/z$) in each retention time (RT) frame is stored as a sparse matrix sized at $N \times W$, where $W$ is the number of ion mobility (IM) bins determined by the IM window width and resolution. It equals the isolation window frame scan range divided by the smallest IM difference between two signals. $N$ is the number of m/z bins, which is determined by the isolation window width and mass resolution. It equals the isolation window m/z range divided by the smallest m/z difference between two signals. To enhance the signal and reduce noise interference[24], data in the same isolation window of adjacent neighbor RT frames are aggregated to form a composite 3D matrix by summing the intensity of signals and subsequently removing the data points lacking proximate neighbors within a certain range. A 2D Gaussian filter is then employed to smooth the $N \times W$ matrix data. Local maxima are identified within the matrix and used as seeds for Gaussian fitting to determine 2D peaks. The center of the 2D Gaussian peak determines the m/z and IM values of the corresponding feature.

In diaTracer, several attributes are defined for each extracted signal, including the m/z and IM centers, as well as the m/z and IM signal ranges. Subsequently, these attributes are partially initialized based on the 2D Gaussian distribution. The sum of the intensities within the signal range represents the peak intensity of the current RT frame. While traversing the frames along the retention time, the algorithm determines the apex for every signal. Upon locating the apex, its signal attributes are finalized, and the intensities from the other RT frames are adjusted relative to it. Signal boundaries are determined based on predefined criteria to ensure a bell-shaped extracted ion chromatogram (XIC). However, if a defined endpoint of the peak is missed, an extended signal is recorded. diaTracer then applies Savitzky-Golay smoothing and the Z-Score[54] peak detection algorithm to segment an elongated peak into multiple separated peaks.

To separate authentic signals from background noise and assign charge states to precursors, diaTracer employs an isotope peak-grouping technique for the MS1 data. The traced MS1 peaks are organized into isotopic clusters based on RT and IM tolerances. The minimum number of the isotopes is 1. The isotope mass difference is based on the number of C13 atoms. Peaks that do not belong to any cluster are discarded. We consider the correlation of the theoretical and experimental isotope intensity distributions, and the correlation of the isotopes' XICs as metrics to control the quality of isotopic clusters. To minimize redundancy, highly confident isotopic peaks passing the threshold are removed from subsequent isotopic clustering. The charge state is determined by comparing against the theoretical isotopic distribution. When the charge state cannot be confidently determined, multiple values are retained. The extraction of MS2 peaks follows a similar process but with more relaxed constraints, including reduced neighbor requirements and no isotopic clustering.

After detecting precursor and fragment XIC features, they are grouped into clusters. Additionally, an optional fractional mass filter is applied to remove peptide features with a fractional mass outside of the expected mass range for tryptic peptides, as previously described[13] ("Mass Defect Filter" parameter). However, this filter is not recommended for certain PTM searches, such as phosphopeptide searches, or nonspecific searches. Clustering begins with the precursor peaks, considering all fragment peaks that met the IM and RT tolerances within the same isolation window ("Delta Apex IM" default 0.02 and "Delta Apex RT" parameters default: 3). Subsequently, the Pearson correlation coefficient is calculated between the smoothed XIC of the monoisotopic precursor and raw XICs of the fragments. Only fragment peaks that surpass the correlation threshold ("Corr threshold" parameter, default 0.3) are included in the construction of the pseudo-MS/MS spectrum. To control the number of fragment peaks in the pseudo-MS/MS spectrum, only the top N highest intensity peaks ("RF max" parameter, default: 500) are included. These default parameters were optimized based on datasets from various LC-MS/MS settings and perform well in most cases. The intensity values in the pseudo-MS/MS spectrum are derived from the peak intensities of the fragment XICs. All produced pseudo-MS/MS spectra are used to generate an mzML file, which can be processed by MSFragger in FragPipe or any other search engine.

### Deep proteome profiling triple-negative breast cancer data analysis

The triple-negative breast cancer (TNBC) dataset from Lapcik et al.[38] contains 12 ddaPASEF runs from a fractionated pooled sample used to generate a DDA-based library and 16 diaPASEF runs from 16 individual TNBC peptide samples. The data was acquired using timsTOF Pro mass spectrometer. We ran FragPipe 22 (pre-release version) with the "DIA_SpecLib_Quant_diaPASEF" workflow to generate a spectral library and perform quantification for the raw .d files. Within diaTracer, "Delta

Apex IM" was set to 0.01, "Delta Apex RT" was set to 3, "RF max" was set to 500, and "Corr threshold" was set to 0.3. The mass defect filter was enabled. In the MSFragger database search, the initial precursor and fragment mass tolerances were set to 10 ppm and 20 ppm, respectively. Spectrum deisotoping[55], mass calibration, and parameter optimization[26] were enabled. The isotope error was set to "0/1/2". The reviewed *Homo sapiens* protein sequence database obtained from UniProt (downloaded on November 15, 2023; 20,461 proteins), appended with common contaminants and decoys, was used in the search. Enzyme specificity was set to "stricttrypsin" and the maximum allowed missed cleavages were set to 2. Oxidation of methionine and N-terminal acetylation were set as variable modifications. The maximum number of variable modifications for each peptide was set to 3. The following change was made compared to the default settings of the "DIA_SpecLib_Quant_diaPASEF" workflow: pyro-Glu at peptide N-terminus was added as a variable modification and methylthiolation of cysteine was used as a fixed modification. MSBooster and Percolator were used to predict the RT and MS/MS spectra, and to rescore PSMs. The final FDR-filtered PSMs and the pseudo-MS/MS mzML spectral files (and DDA mzML files in the hybrid workflow) were used by EasyPQP to generate the spectral library. The spectral library was then used with the DIA-NN quantification module to quantify the library peptide ions in the individual diaPASEF data. The same settings were employed for all experiments, unless mentioned otherwise. Spectronaut 18.5 directDIA result and DIA-NN 1.8.1 library-free result were downloaded from the original study. The Spectronaut report file was exported using Spectronaut Viewer 18.5. DIA-NN generated report.tsv files were processed using the *iq* R package[56] to count the number of peptides and proteins identified passing the specified confidence thresholds (see Result processing and statistical analysis).

### Cerebrospinal fluid data analysis

The cerebrospinal fluid (CSF) dataset from Mun et al.[39] contains 24 ddaPASEF runs used to generate a DDA-based library and 34 diaPASEF runs from 15 patients with Alzheimer's disease (AD) and 19 control subjects. FragPipe with diaTracer was run using the built-in "DIA_SpecLib_Quant_diaPASEF" workflow. Compared to the parameters for the TNBC dataset, carbamidomethylation of cystine set as a fixed modification. For semi-tryptic searches, the cleavage parameter was adjusted to "SEMI". The peptide length was set from 7 to 50 with a peptide mass range of 500–5000 Da. For comparison, DIA-NN (version 1.8.1) as a standalone tool was run in the library-free mode using default settings. The precursor m/z range was set 300–1800. Methionine oxidation was set as a variable modification. Carbamidomethylation of cysteine was used as a fixed modification. A maximum of 1 missed cleavage was allowed. An *in-silico* predicted spectral library was generated from the same database file as described above except without adding decoys. Quantification results from FragPipe and DIA-NN 1.8.1 were processed using the *iq* R package[56] to count the number of peptides and proteins identified passing the specified confidence thresholds.

To conduct comprehensive PTM searches using the mass-offset and open-search mode of MSFragger, FragPipe's "Mass-Offset-CommonPTMs" and "Open" workflows were used under default settings. The details regarding the mass-offset and open search workflows in FragPipe can be found elsewhere[26–28,57]. In brief, in the mass-offset search, MSFragger takes a list of modification masses as mass offsets and searches the spectrum against peptides within a narrow mass tolerance window placed around each specified mass offset. This approach can be used to sensitively and rapidly detect hundreds of known modifications. In the open search mode, MSFragger searches the spectrum against the peptides with a wide mass tolerance window, by default from -150 to 500 Da. It is an efficient approach for detecting peptides with any mass shift, including unknown modifications.

## Plasma data analysis

The subset of the plasma dataset from Vitko et al.[40] used in this study contained 40 files (20 patients with stage IV non-small cell lung cancer, NSCLC, and 20 control samples) acquired using the Seer Proteograph nanoparticle conjugation kit NP2 and a timsTOF HT mass spectrometer. Data was processed as described for the CSF dataset. Differential expression analysis was performed using FragPipe-Analyst[36]. When loading data in FragPipe-Analyst, the "data type" parameter was set to DIA, and both "report.pg_matrix.tsv" and "experiment_annotation.tsv" files were uploaded to the FragPipe-Analyst website. The "min percentage of non-missing values globally" and "Min percentage of non-missing values in at least one condition" parameters were set to 25%, with all other parameters at default values. "DE Adjusted *p*-value cutoff" was set to 0.05. "DE Log2 fold change cutoff" was set to 1. The imputation type was set to "Perseus-type". The FDR correction type was set to "Benjamini Hochberg". The "DE_result.tsv" file was downloaded and used for downstream analysis.

## HLA data analysis

We used a subset of the HLA dataset from Wahle et al.[43] using a sample from one donor profiled in triplicate. FragPipe with diaTracer was run using the built-in "Nonspecific-HLA-diaPASEF" workflow. Compared to the "DIA_SpecLib_Quant_diaPASEF" workflow described above, the enzyme cleavage was set to "nonspecific" and the peptide length was restricted to 7–25. Carbamidomethylation of cysteine was not specified. Cysteinylation (+119) was added as a variable modification. The isotope error was set to 0/1. NetMHCpan 4.1[44] was used to predict the binding affinities of the identified peptides (considering peptides of length 8–12 only). Six MHC class I alleles of the donor were provided as additional input information. The immunopeptides were ranked according to their major histocompatibility complex (MHC)-binding affinities expressed as a percentile rank. Immunopeptides with a percentage rank of <0.5% were classified as strong binders, whereas those with a percentage rank of <2% were considered weak binders. PDV[35], integrated into FragPipe, was used to visualize the PSMs. Both the predicted spectrum and spectral entropy scores were generated using MSBooster[21].

## Phosphoproteome data analysis

The phosphopeptide enriched dataset from Oliinyk et al.[45] contained runs from six gradient lengths ranging from 7 to 60 min. Each gradient condition was replicated four times. To analyze the dataset using FragPipe with diaTracer, we employed the built-in "DIA_SpecLib_Quant_Phospho_diaPASEF" workflow. Compared to the "DIA_SpecLib_Quant_diaPASEF" workflow described above, Phosphorylation on STY was added as a variable modification in MSFragger, and site localization with PTMProphet was enabled. FragPipe propagated the site localization information from the PSM.tsv files to the quantification files produced by DIA-NN ("report.tsv" and "report_pr_matrix.tsv" files). Phosphopeptide counts, representing the number of non-redundant sequences of all identified phosphopeptides, were obtained from FDR-filtered results. When counting the number of phosphorylation sites, sites with the localization probability >0.75 were considered as localized. The quantification intensity correlation shown was generated by R package protti using phosphorylated peptides with the localization probability >0.75 and measured in all 4 technical replicates of the 7 min gradient time experiment. The PSMs for selected peptides were visualized using FragPipe-PDV, whereas XICs were visualized using Skyline[58].

## Low-input, special proteomics data

We used the dataset from Anuar et al.[46] 148 microregions from a patient's tonsil sections. In addition, there were 42 diaPASEF runs (from seven different sizes of human tonsil tissue, six replicates each) that were used to build a comprehensive tonsil library. FragPipe with

diaTracer results were compared to the DIA-NN results from the original study. Differential expression analysis was performed using FragPipe-Analyst based on "report.pg_matrix.tsv" and "experiment_annotation.tsv" files generated by FragPipe. The "min percentage of non-missing values globally" and "Min percentage of non-missing values in at least one condition" parameters were set to 0 and 70%, respectively, with all other parameters at default values. "DE Adjusted *p*-value cutoff" was set to 0.05. "DE Log2 fold change cutoff" was set to 1. The imputation type was set to "Perseus-type". The FDR correction type was set to "Benjamini Hochberg".

## Result processing and statistical analysis

To ensure a fair comparison of peptide and protein identification numbers, DIA-NN "report.tsv" files from the diaTracer workflows in FragPipe and from DIA-NN stand-alone analyses were filtered using the *iq* R package[56] to achieve a 1% FDR at run-specific precursor, global precursor, and global protein levels. Differential expression analyses conducted using FragPipe-Analyst were based on DIA-NN's "report_pg_matrix.tsv" files. Subsequent result processing and plot generation were performed within the RStudio build 402 environment using the R version 4.3.3 statistical software. The R packages ggplot2, tidyverse, ggrepel, plotly, eulerr, and protti were used in our analysis.

## Runtime comparison

The runtimes for FragPipe with diaTracer and for the DIA-NN library-free mode analyses were measured on a Linux desktop with an Intel Core i9-13900K CPU (32 logical cores) and 128GB memory. All analyses were configured to utilize all 32 CPUs.

## Reporting summary

Further information on research design is available in the Nature Portfolio Reporting Summary linked to this article.

## Data availability

The raw MS/MS files used in this study can be found at the ProteomeXchange Consortium and PRIDE partner repository[59] or at the MassIVE repository with the following accession codes: TNBC data: PXD047793. CSF data: PXD035249. Plasma data: PXD047839. HLA data: MSV000092557. Phosphoproteomics data: PXD033904. Low-input, spatial proteomics data: PXD042367. The diaTracer converted mzML files and FragPipe results generated in this study have been deposited in the MassIVE repository with the identifier MSV000094803. Source data are provided with this paper.

## Code availability

The standalone version of diaTracer can be downloaded as a single JAR file at https://diatracer.nesvilab.org. FragPipe is available at GitHub https://github.com/Nesvilab/FragPipe. Python and R scripts for summarizing the results and generating the figures are available at https://github.com/nesvilab/diaTracer-manuscript[60].

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

## Acknowledgements
This work was supported in part by National Institutes of Health grants R01-GM-094231 and U24-CA271037. We thank Maria Wahle for providing additional results files for the HLA dataset.

## Author contributions
K.L. and A.I.N. developed the diaTracer algorithm. K.L. wrote the software and analyzed the results. F.Y. and G.C.T. assisted with the algorithm and software development. K.L.Y helped modify MSBooster. F.Y. assisted with the integration of diaTracer into FragPipe. K.L., F.Y., and A.I.N. wrote the manuscript. A.I.N. conceived the study. A.I.N. and F.Y. supervised the study.

## Competing interests
A.I.N. and F.Y. receive royalties from the University of Michigan for the sale of MSFragger, IonQuant, and diaTracer software licenses to commercial entities. K.L. receives royalties from the University of Michigan for the sale of diaTracer software licenses to commercial entities. All license transactions are managed by the University of Michigan Innovation Partnerships office, and all proceeds are subject to university technology transfer policy. Other authors declare no other competing interests.
