## [Transparent Peer Review file · Nature Communications]

diaTracer enables spectrum-centric analysis of diaPASEF proteomics data

Corresponding Author: Professor Alexey Nesvizhskii

Version 0:

Reviewer comments:

Reviewer #1

(Remarks to the Author)

First, we would like to congratulate the authors on the development and implementation of diaTracer. This represents a significant advance, in particular for TimsTOF users, and adds a complementary tool into the toolbox of DIA analysis workflows. We have tested the software with our own samples and are satisfied with the implementation and results. We believe the paper could be improved in a number of areas, both with respect to the level of detail describing certain aspects of the tool and also how it is compared to alternative search strategies (e.g. using tools such as DIANN, Spectronaut, AlphaPeptDeep).

- When comparing search speed, authors should consider indicating how much of DIANN's search time was dedicated to library generation (Fig2), as this step is not required for every subsequent analysis (similarly to the diaTracer mzml generation)
- The comparison of semi-tryptic and fully tryptic peptides would be enhanced by a peptide level analysis of the 2 selected proteins (or all proteins, Fig3) to reveal whether quantitative differences are due to semi-tryptic peptides or all peptides identified for each protein.
- In connection to the above point, consider including a Woods' plot to indicate fold changes for each peptide, indicating whether they are tryptic or semi-tryptic. While not within the scope of this paper, it may help illustrate how diaTracer can enable more detailed peptide level analysis, potentially revealing important in-vivo derived protease cleavage products.
- The authors only compare the diaTracer tool to the directDIA analysis performed by Wahle et al (Fig4). However, in the original paper, Wahle et. al. shows that this approach is the worst performing and was outperformed by both DDA acquisition and DIA search with predicted library. The authors should at least compare the diaTracer tool to the results obtained with DDA and panlibrary workflows by Wahle et al. It would be beneficial if the tool was compared to all workflows tested by Wahle et al.
- It is nice that the authors show charge state and size distribution on peptide IDs. However, the same analysis should be carried out for the data obtained with the workflows from Wahle et al to investigate if the different computational tools are biased towards certain charge states/peptide sizes.
- Maria et al should be changes to Wahle et al in the figure label (Fig4)
- The gradient-dependent increase in phosphopeptide IDs shown in the supplement is quite interesting and may provide evidence of an advantage of the diaTracer tool over other search strategies. Since it is discussed briefly in the text, authors should consider placing this comparison in the main figure.
- We attempted our own analysis of fresh vs FFPE tissue and observe a very convincing increase in formaldehyde-induced modifications detected by open-search of diaTracer generated mzml files. Without a priori knowledge of the abundant PTMs, it would be impossible, or extremely computationally challenging to observe such a result with other DIA analysis tools. We suggest the authors attempt an offset-search or open-search analysis that can illustrate the power of their tool to reveal PTM differences across different sample conditions. The paper currently only provides an example that an open search can be done, but not strong evidence of its validity or potential.
- Authors should add a description of the parameters for diaTracer mzml generation (i.e. 3D precursor isolation) and how these affect performance. It is not clear whether the settings for Delta Apex IM, Delta Apex RT, RF Max, Corr threshold are optimized based on the data itself during pseudo-MS2 generation, or whether these settings should be adjusted by the user based on their specific chromatography or instrument IM acquisition settings.

The paper itself is clear, well written and scientifically sound.

(Remarks on code availability)

We have both installed and used the application. While we did not reproduce the specific analyses contained within the manuscript, we are confident of the ability for the tool to perform as described (generate pseudo MS2 .mzml spectra from Bruker .d diaPASEF files, and perform tryptic, semi-tryptic or open searches on the resulting files).

Reviewer #2

(Remarks to the Author)

Main reviewer's comments:

The diaTracer tool deconvolutes diaPASEF data into DDA-like pseudo-MS/MS spectra. Integrated into FragPipe, diaTracer facilitates comprehensive downstream analysis, including peptide identification, deep-learning-based rescoring, protein inference, FDR filtering, PTM site localization, spectral library generation, quantification, and data visualization. The diaTracer-FragPipe workflow is applied to five applications: CSF, plasma, HLA, phospho, and low-input samples. This workflow demonstrates significant advantages for analyzing diaPASEF datasets with large search spaces, which are impractical for peptide-centric strategies. The work is notable, and considering the following comments could further enhance the manuscript.

1. Please rephrase the below sentence.

87 is that it facilitates analysis

88 if data in experiments that require a very large search space.

2. diaTracer-FragPipe workflow is mainly compared to DIA-NN (e.g. in the CSF, plasma, and low-input samples), but not Spectronaut. What is the reason for it?

95 Although Spectronaut supports

96 direct identification from diaPASEF data to the best of our knowledge, the details of the algorithm

97 have not been publicly disclosed.

3. The meaning of mass-offset search is not clear. It seems that it is a special search mode for PTM.

136 mass-offset, or even open database searches

162 comprehensive PTM searches (using the mass offset mode of MSFragger).

4. Mass-offset searches took 341 min, from MSFragger search to PTM-Shepherd reports. In the below sentences, are you comparing the time cost of mass-offset search to open search? Many details are missing in the second half sentence, as nothing about open search is discussed previously.

209 Similar mass shift histogram was observed in the open search, albeit at a 2x

210 longer computational analysis time.

5. In the below sentences, how do the semi-tryptic peptides of AKT2 and H2AX affect their protein quantification? The explanation will further demonstrate the importance of semi-tryptic search.

239 Figure 3d shows two examples of proteins showing statistically

240 significant upregulation in the NSCLC vs. control group with the semi-tryptic search, for which a

241 non-significant difference was observed with the tryptic search. These two proteins, AKT2 and

242 H2AX, have been previously reported to be associated with NSCLC.

6. Please rephrase the below sentence ("beneficial", "required" are not clear).

375 Even semi-tryptic searches that are often beneficial (e.g., CSF and plasma samples)

376 or required (N-terminomics) are challenging.

7. Please make 'the accuracy of quantification' clearer.

386 We demonstrated that the semi-enzymatic search provides a significant

387 boost in the number of quantified precursors and improves the accuracy of quantification in the

388 analysis of CSF and plasma datasets.

8. diaTracer algorithm focuses on feature detection (from original data to pseudo-MS/MS spectra and mzML files). diaTracer-FragPipe workflow includes diaTracer module, and other downstream data analysis (e.g. identification, filtering, quantification). Please make it clear that here is diaTracer module algorithm (diaTracer-FragPipe workflow is described previously: 116 diaTracer workflow in FragPipe).

419 diaTracer algorithm

9. The working mechanism of the "Mass Defect Filter" is not clearly explained, since it has been mentioned several times.

453 mass defect filter ("Mass Defect Filter" parameter) can be employed to reduce the number of 454 peaks before clustering without any impact on the outcomes of most conventional searches

10. 's localized' is 'as localized'

539 localization probability greater than 0.75 were considered s localized.

Co-reviewer's comments:

1. Regarding the timsTOF data structure, there are concepts of 'Frame' and 'Scan.' The 'Frame' corresponds to the RT scale, while the 'Scan' corresponds to the ion mobility scale. This differs from the traditional MS data where 'Scan' corresponds to the RT scale. In the methods section, the authors only use 'Scan' to represent the MS data. It is unclear whether they are referring to the Scan at the RT scale or the ion mobility scale.
2. The authors mention dividing the peaks into m/z bins and IM bins based on window width and resolution. Could they elaborate on how this was done in detail? Specifically, how are the bins determined by window width and resolution?
3. The authors state that they aggregated adjacent neighbor scans. First, it is unclear whether this aggregation was done at the RT scale or the IM scale, as mentioned in the first comment. Second, it is unclear how they aggregated the signal. Did they use the maximum or the sum of the signal from neighboring scans?
4. The authors state that "a 2D Gaussian filter is then employed to smooth the composite 3D matrix data." How is this performed? Is the filter first applied to the m/z scale and then to the IM scale? Is the data smoothed twice? Have the authors considered identifying 3D features directly rather than finding 2D Gaussian-based features in each dimension separately?
5. When organizing MS1 peaks into isotopic clusters, what is the minimum number of isotopes required for a precursor? Is the isotope mass based solely on the number of C13 atoms, or does it include both C13 and N15 atoms?
6. In line 445, the authors state: "To minimize redundancy, highly confident peaks are removed from subsequent isotopic clustering." How are "highly confident peaks" defined? What are the criteria?
7. Each feature should have both a Chromatogram (XIC) and a Mobilogram. It seems the authors only considered Chromatograms in their method. When clustering fragments with precursors, would considering the correlation of both chromatograms and mobilograms yield better results?
8. In line 460, the authors state: "Only fragment peaks that surpass the correlation threshold ('Corr threshold' parameter) are included in the construction of the pseudo-spectrum." What is the minimal possible cutoff for this 'Corr threshold' parameter? Must it be greater than 0.5 for peaks to be considered correlated?
9. The pseudo-MS/MS spectrum method may offer an advantage by reducing the search space. However, it also comes with a limitation: it relies on good MS1 features to generate the pseudo spectrum. In contrast, peptide-centric approaches primarily rely on MS2 XICs for identification, with MS1-level XIC being optional. Consequently, the pseudo-MS/MS spectrum method may be unable to detect precursors that do not have strong MS1-level XICs. Have the authors considered any solutions for this issue?
10. The authors extract XICs during feature identification, but later discard this information, retaining only the Spectrum information in the spectral library and using DIA-NN to extract the XICs again for quantification. Have they considered reusing the initial XICs for quantification?
11. How did the authors use DIA-NN solely for quantification? To my understanding, DIA-NN always performs an identification step first when starting from a spectral library. If the authors claim they only used DIA-NN for quantification, what additional parameters were used? Do they need to use the FDR cutoff calculated by DIA-NN? Have they considered developing their own quantification method rather than relying on DIA-NN?

(Remarks on code availability)

The URL (<https://github.com/Nesvilab/diaTracer>) provides the README file with the instructions for installing and running test data. Instructions from the URL (<https://github.com/Nesvilab/diaTracer>) are not working in my laptop.

Instructions from "510296_0_related_ms_9081585_sdsqtv.docx" are working (as below).

Instructions to run on data.

1. Download FragPipe from <https://github.com/Nesvilab/FragPipe/releases/latest>
2. Configure FragPipe following the instructions on https://fragpipe.nesvilab.org/docs/tutorial_setup_fragpipe.html and https://fragpipe.nesvilab.org/docs/tutorial_fragpipe.html#configure-fragpipe.
3. Download the demo data from https://www.dropbox.com/scl/fi/ydoeyjcxue6yz3c4rthvc/demo_data.zip?rlkey=5hazgmw4qfdc25l8oco5z3s8q&dl=1
Unzip it.
4. In the FragPipe "Workflow" tab, load the "DIA_Speclib_Quant_diaPASEF" workflow.
5. In the FragPipe "Workflow" tab, load the "20200505_Evosep_100SPD_SG06-16_MLHeLa_100ng_py8_S2-C1_1_2731.d" file in the "demo_data" folder and assign it to "DIA" data type.
6. In the FragPipe "Database" tab, load the fasta file "2023-11-15-decoys-2023-11-15-reviewed-contam-UP000005640.fas" in the "demo_data" folder.
7. In the FragPipe "Run" tab, specify the output directory, and click "RUN".

Expected output:

A mzML file named "20200505_Evosep_100SPD_SG06-16_MLHeLa_100ng_py8_S2-C1_1_2731_diaTracer.mzML" will be generated in the same folder of input "20200505_Evosep_100SPD_SG06-16_MLHeLa_100ng_py8_S2-C1_1_2731.d" file.

Many intermediate and result files are generated by FragPipe and DIA-NN in the output directory. For example, "psm.tsv" and "protein.tsv" file contains all precursors and proteins respectively identified by FragPipe; a spectral library file "library.tsv"; all quantification result files in "diann-output" folder.

Expected run time on a 12 threads desktop computer:
15 minutes.

Reviewer #3

(Remarks to the Author)

(Remarks on code availability)

The link <https://github.com/Nesvilab/diaTracer-manuscript> dose not work. Also, they didn't provide the source code of the algorithm.

Reviewer #4

(Remarks to the Author)

I co-reviewed this manuscript with one of the reviewers who provided the listed reports. This is part of the Nature Communications initiative to facilitate training in peer review and to provide appropriate recognition for Early Career Researchers who co-review manuscripts

(Remarks on code availability)

Code is not available

Reviewer #5

(Remarks to the Author)

In this work, the authors present diaTracer for the spectrum-centric analysis of diaPASEF data. The diaTracer generates pseudo-MS/MS spectra based on XIC features between all detected precursors and fragments within user-defined ion mobility and retention time tolerance. Then the generated pseudo-MS/MS spectra were searched by FragPipe for further analysis, such as open search, PTMs analysis, etc. Examples are provided for the analysis of CSF data, plasma proteome, HLA immunopeptides, phosphopeptides, low-input and spatial proteomics.

Overall, the authors provide a nice tool in spectrum-centric analysis of DIA data. The main criticism is about the novelty of the work. The idea of diaTracer is an supplementary of DIA-Umpire, which was developed by the same group. The main difference is taking ion mobility into consideration by diaTracer. DIA-Umpire considers the elution profiles similarity to group precursors and fragments and then generates pseudo-MS/MS spectra. The diaTracer considers additionally the ion mobility information. After the generation of the pseudo-MS/MS spectra, all the analysis is the same as the standard FragPipe workflow. In view of this, the novelty of the work is doubtful for publication on Nature Communications, which normally publish work with significant novelty.

In addition, there are some technique issues:

- The authors mentioned that Spectronaut supports direct identification from diaPASEF data. Although the algorithm is not disclosed, the tool should be compared with the newest version of Spectronaut 19 for benchmarking purpose. For instance, by analyzing the CSF data mentioned in this work, 1361 protein groups can be identified in average or 1600 in total using the directDIA mode of Spectronaut 19 (not the hybrid library mode, which can generate even more identification), more than the number of identifications discussed in this submission. Besides, the version of Spectronaut which authors mentioned in HLA immunopeptidomics datasets is too old.
- Clinical samples are not suitable for benchmarking new tools. The authors should start from simple datasets like HeLa. The authors should further explain the purpose of selection of these datasets. Is there an internal logic to these selections?
- The authors compared diaTracer with other works before. In most other works, they used DIA-NN originally. The authors need to check the version and parameters to ensure that they are consistent.
- The accuracy should be compared in addition to the numbers of identification, for instance, the consistence between the spectrum-centric analysis and peptide-centric analysis of diaPASEF data, and the consistence with other software solutions, e.g. DIA-NN and Spectronaut. The accuracy and precision in quantification should also be compared.

- In Methods, the authors should provide precise arithmetic formulas explaining the diaTracer algorithms to make the description more explicit. For example, the authors mentioned "data from adjacent neighbor scans are aggregated to form a composite 3D matrix, subsequently removing the data points lacking proximate neighbors.". Firstly, the definition of distance should be provided in arithmetic formulas. Secondly, it doesn't make sense when the authors mentioned "data points lacking proximate neighbors" as apparently all data points have proximate neighbors and what does matter in this scenario is the distance between data points and their proximate neighbors. Vague description like prementioned should be demonstrated more explicitly with arithmetic formulas.

Minor opinions:

- Way too much bar plots are employed in the article, please use other ways to show the results.

(Remarks on code availability)

The codes are clear.

Version 1:

Reviewer comments:

Reviewer #1

(Remarks to the Author)

We are satisfied by the detailed responses from the authors to our comments and recommend the paper be published in Nature Communications.

(Remarks on code availability)

We have both installed and used the application. While we did not reproduce the specific analyses contained within the manuscript, we are confident of the ability for the tool to perform as described (generate pseudo MS2 .mzml spectra from Bruker .d diaPASEF files, and perform tryptic, semi-tryptic or open searches on the resulting files).

Reviewer #2

(Remarks to the Author)

The authors addressed all my questions. No further questions. Thanks.

(Remarks on code availability)

I installed and tested the example.

Reviewer #3

(Remarks to the Author)

(Remarks on code availability)

Since they are not open-source, the code dose not contain any algorithm details.

Reviewer #5

(Remarks to the Author)

The authors have made some improvement by this revision. However, the main concerns are not fully addressed.

It is appreciated that the authors develop open-source or free software solutions for proteomics, which benefits the society. However, to publish an article on distinguished journals like Nature Communications, I believe extensive benchmarking analysis is required. The authors argued that Spectronaut is not free and doesn't publish their algorithm for diaPASEF analysis, which are true. Nevertheless, it is a software widely used for proteomics, including in the analysis of diaPASEF data. There is no evidence that the results from Spectronaut are not reliable. In view of this, I believe extensive comparison with Spectronaut is required to convince people using diaTracer. As the authors do not have access to Spectronaut software, I would suggest to perform more comparison with the published results using Spectronaut. A single comparison on one dataset can take bias. I have quickly searched through the diaPASEF datasets with Spectronaut directDIA analysis results public available, and found <https://doi.org/10.1038/s41597-024-03632-2> with Spectronaut 18.5 and <https://pubs.acs.org/doi/10.1021/acs.jproteome.2c00735> with Spectronaut 17.

For the accuracy in quantification, the authors added the new analysis results in Figure 5b. However, there is a lack of comparison of the results with those from other analysis methods, including the spectral library based FragPipeDIA and DIA-NN, as well as Spectronaut. Coefficient of variation should be compared.

(Remarks on code availability)

The authors only provided codes to reproduce the figures. diaTracer is not open source. The reviewer cannot further validate the software.

Version 2:

Reviewer comments:

Reviewer #5

(Remarks to the Author)

The authors have addressed all my concerns. I have no further questions.

(Remarks on code availability)

I accept the explanation by the authors on the code availability.

REVIEWER COMMENTS

Reviewer #1 (Remarks to the Author):

First, we would like to congratulate the authors on the development and implementation of diaTracer. This represents a significant advance, in particular for TimsTOF users, and adds a complementary tool into the toolbox of DIA analysis workflows. We have tested the software with our own samples and are satisfied with the implementation and results. We believe the paper could be improved in a number of areas, both with respect to the level of detail describing certain aspects of the tool and also how it is compared to alternative search strategies (e.g. using tools such as DIANN, Spectronaut, AlphaPeptDeep).

Response: We thank the reviewer for the kind words and positive feedback. We have addressed all the comments. Please find our point-by-point responses below.

- When comparing search speed, authors should consider indicating how much of DIANN's search time was dedicated to library generation (Fig2), as this step is not required for every subsequent analysis (similarly to the diaTracer mzml generation)

Response: We have added the information regarding the DIA-NN spectral library generation step to the main text.

In contrast, the DIA-NN library-free analysis took 10.5 min to generate the in-silico predicted spectral library and 1,486 min to perform the rest of the analysis, more than twice the time of the diaTracer workflow. Importantly, repeating FragPipe analysis (starting with the existing pseudo-MS/MS mzML files) using the semi-tryptic search took only 110 min. Because diaTracer extracts

all possible signals, the generated pseudo-MS/MS mzML file can be reused in any search setting. In contrast, the in-silico spectral library generated by DIA-NN can only be reused if the search settings are the same.

- The comparison of semi-tryptic and fully tryptic peptides would be enhanced by a peptide level analysis of the 2 selected proteins (or all proteins, Fig3) to reveal whether quantitative differences are due to semi-tryptic peptides or all peptides identified for each protein.

Response: We thank the reviewer for the suggestion. We examined the peptides of these two proteins, see Supplementary Figure 4. We also added the following description to the text.

Using the semi-tryptic search setting, we identified 30,158 peptides, compared to 26,469 peptides identified using the tryptic search. Of these, 7,642 peptides were uniquely identified in the semi-tryptic search (Supplementary Figure 3). Among these 7,642 peptides, 6,800 (89%) were semi-tryptic. Additionally, 95% (6,458) of the semi-tryptic peptides, corresponding to 1,104 proteins, were quantified.

Upon examining peptide-level quantification, we found the difference in H2AX quantification was due to two semi-tryptic peptides showing significant upregulation in the NSCLC group (Supplementary Figure 4). For AKT2, however, there were no semi-tryptic peptides quantified. One tryptic peptide identified in the tryptic search was missing from the semi-tryptic search as it did not pass the FDR filters. In the absence of this (low confidence) peptide, AKT2 expression was significantly upregulated in the NSCLC group.

- In connection to the above point, consider including a Woods' plot to indicate fold changes for each peptide, indicating whether they are tryptic or semi-tryptic. While not within the scope of this paper, it may help illustrate how diaTracer can enable more detailed peptide level analysis, potentially revealing important in-vivo derived protease cleavage products.

Response: We thank the reviewer for the suggestion. The Woods' plot for H2AX is shown in Supplementary Figure 4.

- The authors only compare the diaTracer tool to the directDIA analysis performed by Wahle et al (Fig4). However, in the original paper, Wahle et. al. shows that this approach is the worst performing and was outperformed by both DDA acquisition and DIA search with predicted library. The authors should at least compare the diaTracer tool to the results obtained with DDA and panlibrary workflows by Wahle et el. It would be beneficial if the tool was compared to all workflows tested by Wahle et al.

Response: We added several new comparisons (Supplementary Figure 6) and revised the text accordingly.

We also compared how the identified immunopeptides overlapped with those reported in the original study using the experimental DDA library and using the panlibrary (Supplementary Figure 6). While it is expected that the analysis of diaPASEF data using a more comprehensive spectral library would result in more peptide identifications, our FP-diaTracer results showed good overlap with the published data.

- It is nice that the authors show charge state and size distribution on peptide IDs. However, the same analysis should be carried out for the data obtained with the workflows from Wahle et al to investigate if the different computational tools are biased towards certain charge states/peptide sizes.

Response: We thank the reviewer for the suggestion. We have added the plots (Figure 4c and 4d).

- Maria et al should be changes to Wahle et al in the figure label (Fig4)

Response: It has been corrected.

- The gradient-dependent increase in phosphopeptide IDs shown in the supplement is quite interesting and may provide evidence of an advantage of the diaTracer tool over other search strategies. Since it is discussed briefly in the text, authors should consider placing this comparison in the main figure.

Response: We thank the reviewer for the suggestion. We have included this information in the main figure (Figure 5a).

- We attempted our own analysis of fresh vs FFPE tissue and observe a very convincing increase in formaldehyde-induced modifications detected by open-search of diaTracer generated mzml files. Without a priori knowledge of the abundant PTMs, it would be impossible, or extremely computationally challenging to observe such a result with other DIA analysis tools. We suggest the authors attempt an offset-search or open-search analysis that can illustrate the power of their tool to reveal PTM differences across different sample conditions. The paper currently only provides an example that an open search can be done, but not strong evidence of its validity or potential.

Response: We thank the reviewer for testing diaTracer for open PTM searches and for such positive feedback. We are also very excited about the possibility of doing such

searches. However, we feel there is only so many analyses we can fit in one paper while staying focused, so we would like to leave a detailed analysis of PTM profiles from open searches to future work (or leave it to our users to explore and publish).

- Authors should add a description of the parameters for diaTracer mzml generation (i.e. 3D precursor isolation) and how these affect performance. It is not clear whether the settings for Delta Apex IM, Delta Apex RT, RF Max, Corr threshold are optimized based on the data itself during pseudo-MS2 generation, or whether these settings should be adjusted by the user based on their specific chromatography or instrument IM acquisition settings.

Response: These parameters are not sensitive to sample type or data acquisition parameters. According to our testing, the default settings, except for the mass defect filter, worked well in most cases. For PTM analysis, the mass defect filter should be disabled, as explained in the Methods section. We revised the text to improve the clarity.

After detecting precursor and fragment XIC features, they are grouped into clusters. Additionally, an optional fractional mass filter is applied to remove peptide features with a fractional mass outside of the expected mass range for tryptic peptides, as previously described¹³ ("Mass Defect Filter" parameter). However, this filter is not recommended for certain PTM searches, such as phosphopeptide searches, or nonspecific searches. Clustering begins with the precursor peaks, considering all fragment peaks that met the IM and RT tolerances within the same isolation window ("Delta Apex IM" default 0.02 and "Delta Apex RT" parameters default: 3). Subsequently, the Pearson correlation coefficient is calculated between the smoothed XIC of the monoisotopic precursor and raw XICs of the fragments. Only fragment peaks that surpass the correlation threshold ("Corr threshold" parameter, default 0.3) are included in the construction of the pseudo-MS/MS spectrum. To control the number of fragment peaks in the pseudo-MS/MS spectrum, only the top N highest intensity peaks ("RF max" parameter, default: 500) are included. These default parameters were optimized based on datasets from various LC-MS/MS settings and perform well in most cases.

The paper itself is clear, well written and scientifically sound.

Response: We thank the reviewer for the kind words. We will keep maintaining the diaTracer.

Reviewer #1 (Remarks on code availability):

We have both installed and used the application. While we did not reproduce the specific analyses contained within the manuscript, we are confident of the ability for the tool to perform as described (generate pseudo MS2 .mzml spectra from Bruker .d diaPASEF files, and perform tryptic, semi-tryptic or open searches on the resulting files).

Response: We thank the reviewer for the positive feedback. We are glad to hear that the tool works as described.

Reviewer #2 (Remarks to the Author):

Main reviewer's comments:

The diaTracer tool deconvolutes diaPASEF data into DDA-like pseudo-MS/MS spectra. Integrated into FragPipe, diaTracer facilitates comprehensive downstream analysis, including peptide identification, deep-learning-based rescoring, protein inference, FDR filtering, PTM site localization, spectral library generation, quantification, and data visualization. The diaTracer-FragPipe workflow is applied to five applications: CSF, plasma, HLA, phospho, and low-input samples. This workflow demonstrates significant advantages for analyzing diaPASEF datasets with large search spaces, which are impractical for peptide-centric strategies. The work is notable, and considering the following comments could further enhance the manuscript.

Response: We thank the reviewer for these positive comments.

1. Please rephrase the below sentence.

“is that it facilitates analysis if data in experiments that require a very large search space”.

Response: We have rephrased it:

One important advantage of the spectrum-centric DIA-Umpire strategy of deconvoluting full DIA MS/MS spectra into pseudo-MS/MS spectra is that it facilitates analyses requiring a large search space.

2. diaTracer-FragPipe workflow is mainly compared to DIA-NN (e.g. in the CSF, plasma, and low-input samples), but not Spectronaut. What is the reason for it?

Response: We compared the results with those from Spectronaut when the results were made available as part of the original publication (e.g. Figure 4). For the other experiments, we were not able to perform the comparisons. Spectronaut is a commercial software that requires a paid license, even for academic scientists, to download. We do not have access to it. Furthermore, to the best of our knowledge, the Spectronaut's method for diaPASEF analysis has not been published in peer-reviewed literature (or anywhere in detail, really). Without the full description of their method and without

knowing how the software works, we are not in a position to perform a fair comparison ourselves.

3. The meaning of mass-offset search is not clear. It seems that it is a special search mode for PTM.

Response: We have added additional details and references regarding the mass offset search mode of MSFragger to the text:

To conduct comprehensive PTM searches using the mass-offset and open-search mode of MSFragger, FragPipe's "Mass-Offset-CommonPTMs" and "Open" workflows were used under default settings. The details regarding the mass-offset and open search workflows in FragPipe can be found elsewhere^s. In brief, in the mass-offset search, MSFragger takes a list of modification masses as mass offsets and searches the spectrum against peptides within a narrow mass tolerance window placed around each specified mass offset. This approach can be used to sensitively and rapidly detect hundreds of known modifications. In the open search mode, MSFragger searches the spectrum against the peptides with a wide mass tolerance window, by default from -150 to 500 Da. It is an efficient approach for detecting peptides with any mass shift, including unknown modifications.

4. Mass-offset searches took 341 min, from MSFragger search to PTM-Shepherd reports. In the below sentences, are you comparing the time cost of mass-offset search to open search? Many details are missing in the second half sentence, as nothing about open search is discussed previously.

Similar mass shift histogram was observed in the open search, albeit at a 2x longer computational analysis time.

Response: We have rephrased the sentence and added Supplementary Figure 2 showing more details. We also provided more details in the Methods.

A similar mass shift histogram (Supplementary Figure 2) was observed from the open search results using the FragPipe's "Open" workflow, and the computational analysis took similar time (317 min).

To conduct comprehensive PTM searches using the mass-offset and open-search mode of MSFragger, FragPipe's "Mass-Offset-CommonPTMs" and "Open" workflows were used under default settings. The details regarding the mass-offset and open search workflows in FragPipe can be found elsewhere^{27-29, 58s}. In brief, in the mass-offset search, MSFragger takes a list of modification masses as mass offsets and searches the spectrum against peptides within a narrow mass tolerance window placed around each specified mass offset. This approach can be used to sensitively and rapidly detect hundreds of known modifications. In the open search

mode, MSFragger searches the spectrum against the peptides with a wide mass tolerance window, by default from -150 to 500 Da. It is an efficient approach for detecting peptides with any mass shift, including unknown modifications.

5. In the below sentences, how do the semi-tryptic peptides of AKT2 and H2AX affect their protein quantification? The explanation will further demonstrate the importance of semi-tryptic search.

Figure 3d shows two examples of proteins showing statistically significant upregulation in the NSCLC vs. control group with the semi-tryptic search, for which a non-significant difference was observed with the tryptic search. These two proteins, AKT2 and H2AX, have been previously reported to be associated with NSCLC.

Response: We thank the reviewer for the suggestions. We revised the text and added Supplementary Figures 3 and 4.

Using the semi-tryptic search setting, we identified 30,158 peptides, compared to 26,469 peptides identified using the tryptic search. Of these, 7,642 peptides were uniquely identified in the semi-tryptic search (Supplementary Figure 3). Among these 7,642 peptides, 6,800 (89%) were semi-tryptic. Additionally, 95% (6,458) of the semi-tryptic peptides, corresponding to 1,104 proteins, were quantified.

Upon examining peptide-level quantification, we found the difference in H2AX quantification was due to two semi-tryptic peptides showing significant upregulation in the NSCLC group (Supplementary Figure 4). For AKT2, however, there were no semi-tryptic peptides quantified. One tryptic peptide identified in the tryptic search was missing from the semi-tryptic search as it did not pass the FDR filters. In the absence of this (low confidence) peptide, AKT2 expression was significantly upregulated in the NSCLC group.

6. Please rephrase the below sentence (“beneficial”, “required” are not clear).

“Even semi-tryptic searches that are often beneficial (e.g., CSF and plasma samples) or required (N-terminomics) are challenging.”

Response: This has been rephrased.

Even semi-tryptic searches, which are often beneficial for certain samples, such as plasma samples and samples from N-terminomics^{48, 49} studies, remain challenging.

7. Please make ‘the accuracy of quantification’ clearer.

“We demonstrated that the semi-enzymatic search provides a significant boost in the number of quantified precursors and improves the accuracy of quantification in the analysis of CSF and plasma datasets.”

Response: Accuracy refers to the closeness of the measurement to the underlying true value. This sentence has been revised accordingly.

We demonstrated that the semi-enzymatic search provides a significant boost in the number of quantified precursors. It also increases the accuracy and precision of quantification, thereby identifying more differentially expressed proteins in the analysis of CSF and plasma datasets.

8. diaTracer algorithm focuses on feature detection (from original data to pseudo-MS/MS spectra and mzML files). diaTracer-FragPipe workflow includes diaTracer module, and other downstream data analysis (e.g. identification, filtering, quantification). Please make it clear...

Response: We have renamed the section “diaTracer feature extraction algorithm” and added additional details regarding the algorithm.

The data from one isolation window ($1/k0 \times m/z$) in each retention time (RT) frame is stored as a sparse matrix sized at $N \times W$, where W is the number of ion mobility (IM) bins determined by the IM window width and resolution. It equals the isolation window frame scan range divided by the smallest IM difference between two signals. N is the number of m/z bins, which is determined by the isolation window width and mass resolution. It equals the isolation window m/z range divided by the smallest m/z difference between two signals. To enhance the signal and reduce noise interference²⁵, data in the same isolation window of adjacent neighbor RT frames are aggregated to form a composite 3D matrix by summing the intensity of signals and subsequently removing the data points lacking proximate neighbors within a certain range. A 2D Gaussian filter is then employed to smooth the $N \times W$ matrix data. Local maxima are identified within the matrix and used as seeds for Gaussian fitting to determine 2D peaks. The center of the 2D Gaussian peak determines the m/z and IM values of the corresponding feature.

9. The working mechanism of the "Mass Defect Filter" is not clearly explained, since it has been mentioned several times.

“mass defect filter ("Mass Defect Filter" parameter) can be employed to reduce the number of peaks before clustering without any impact on the outcomes of most conventional searches”

Response: We have revised the text and added citation to our paper explaining the fractional mass filter and how it is used.

After detecting precursor and fragment XIC features, they are grouped into clusters. Additionally, an optional fractional mass filter is applied to remove peptide features with a fractional mass outside of the expected mass range for tryptic peptides, as previously described¹³ ("Mass Defect Filter" parameter). However, this filter is not recommended for certain PTM searches, such as phosphopeptide searches, or nonspecific searches.

10. 's localized' is 'as localized'

Line 539: "localization probability greater than 0.75 were considered s localized."

Response: We have corrected the typo.

Co-reviewer's comments:

1. Regarding the timsTOF data structure, there are concepts of 'Frame' and 'Scan.' The 'Frame' corresponds to the RT scale, while the 'Scan' corresponds to the ion mobility scale. This differs from the traditional MS data where 'Scan' corresponds to the RT scale. In the methods section, the authors only use 'Scan' to represent the MS data. It is unclear whether they are referring to the Scan at the RT scale or the ion mobility scale.

Response: We have rephrased the sentences.

The data from one isolation window ($1/k_0 \times m/z$) in each retention time (RT) frame is stored as a sparse matrix sized at $N \times W$, where W is the number of ion mobility (IM) bins determined by the IM window width and resolution. It equals the isolation window frame scan range divided by the smallest IM difference between two signals. N is the number of m/z bins, which is determined by the isolation window width and mass resolution. It equals the isolation window m/z range divided by the smallest m/z difference between two signals. To enhance the signal and reduce noise interference²⁵, data in the same isolation window of adjacent neighbor RT frames are aggregated to form a composite 3D matrix by summing the intensity of signals and subsequently removing the data points lacking proximate neighbors within a certain range. A 2D Gaussian filter is then employed to smooth the $N \times W$ matrix data. Local maxima are identified within the matrix and used as seeds for Gaussian fitting to determine 2D peaks. The center of the 2D Gaussian peak determines the m/z and IM values of the corresponding feature.

2. The authors mention dividing the peaks into m/z bins and IM bins based on window width and resolution. Could they elaborate on how this was done in detail? Specifically, how are the bins determined by window width and resolution?

Response: We have rephrased the sentences and provided more details, see above.

3. The authors state that they aggregated adjacent neighbor scans. First, it is unclear whether this aggregation was done at the RT scale or the IM scale, as mentioned in the first comment. Second, it is unclear how they aggregated the signal. Did they use the maximum or the sum of the signal from neighboring scans?

Response: We have rephrased the sentences and added more details.

To enhance the signal and reduce noise interference²⁵, data in the same isolation window of adjacent neighbor RT frames are aggregated to form a composite 3D matrix by summing the intensity of signals and subsequently removing the data points lacking proximate neighbors within a certain range.

4. The authors state that "a 2D Gaussian filter is then employed to smooth the composite 3D matrix data." How is this performed? Is the filter first applied to the m/z scale and then to the IM scale? Is the data smoothed twice? Have the authors considered identifying 3D features directly rather than finding 2D Gaussian-based features in each dimension separately?

Response: 2D Gaussian kernel was used. We have clarified it in the text. The data is smoothed only once. We have considered multiple strategies, including 3D feature detection. There are advantages and drawbacks to each strategy. The manuscript describes what we have currently implemented.

5. When organizing MS1 peaks into isotopic clusters, what is the minimum number of isotopes required for a precursor? Is the isotope mass based solely on the number of C13 atoms, or does it include both C13 and N15 atoms?

Response: To maintain the highest sensitivity, the minimum number of isotopes used was 1. It is based solely on the number of C13 atoms. We have clarified in the text.

The minimum number of the isotopes is 1. The isotope mass difference is based on the number of C13 atoms. Peaks that do not belong to any cluster are discarded.

6. In line 445, the authors state: "To minimize redundancy, highly confident peaks are removed from subsequent isotopic clustering." How are "highly confident peaks" defined? What are the criteria?

Response: We have rephrased the sentences and added more details.

We consider the correlation of the theoretical and experimental isotope intensity distributions, and the correlation of the isotopes' XICs as metrics to control the quality of isotopic clusters. To

minimize redundancy, highly confident isotopic peaks passing the threshold are removed from subsequent isotopic clustering.

7. Each feature should have both a Chromatogram (XIC) and a Mobilogram. It seems the authors only considered Chromatograms in their method. When clustering fragments with precursors, would considering the correlation of both chromatograms and mobilograms yield better results?

Response: After signal extraction, we obtained centroided ion mobility. We did not calculate the mobilogram correlation in order to maintain reasonably fast computational speed.

8. In line 460, the authors state: "Only fragment peaks that surpass the correlation threshold ('Corr threshold' parameter) are included in the construction of the pseudo-spectrum." What is the minimal possible cutoff for this 'Corr threshold' parameter? Must it be greater than 0.5 for peaks to be considered correlated?

Response: We used 0.3 by default, because most of the low-abundance fragments did not have good XIC shapes. We have clarified it in the text.

Only fragment peaks that surpass the correlation threshold ("Corr threshold" parameter, default 0.3) are included in the construction of the pseudo-MS/MS spectrum.

9. The pseudo-MS/MS spectrum method may offer an advantage by reducing the search space. However, it also comes with a limitation: it relies on good MS1 features to generate the pseudo spectrum. In contrast, peptide-centric approaches primarily rely on MS2 XICs for identification, with MS1-level XIC being optional. Consequently, the pseudo-MS/MS spectrum method may be unable to detect precursors that do not have strong MS1-level XICs. Have the authors considered any solutions for this issue?

Response: We have considered many strategies to address this issue over the years, going back as far as our early DIA-Umpire work. One need to have a good balance between the sensitivity and the specificity of feature extraction (the latter gets worse when too many features are extracted) and the computational speed. What diaTracer does now is that it aggregates neighboring scans to amplify the signals. It can extract MS1 XICs even if they are weak. However, it may still miss peptides with very weak MS1 signals, and it would not extract features with absolutely no MS1 signal. We have added a new paragraph in Discussion describing the limitations.

The spectrum-centric approach of diaTracer (and DIA-Umpire) also has limitations. First, diaTracer starts with MS1 feature detection, which means it relies on the detection of precursor

peptide's signal in MS1 data to group fragments and build a pseudo-MS/MS spectrum. To minimize this limitation, diaTracer aggregates the neighboring RT frames to amplify the MS1 signals. This helps to extract weak signals and filter out noise. Second, diaTracer extracts multiple features for each signal, including ion mobility range, m/z, precursor XIC, XICs of the corresponding fragments and their correlation with the precursor and with each other. However, this information is currently used only at the pseudo-MS/MS generation step. Additionally using it in the subsequent database search and scoring, as it is done in e.g. MSFragger-DIA¹⁹, would likely improve the results. We plan to address these limitations in future work.

10. The authors extract XICs during feature identification, but later discard this information, retaining only the Spectrum information in the spectral library and using DIA-NN to extract the XICs again for quantification. Have they considered reusing the initial XICs for quantification?

Response: Yes, we certainly considered it, and we do plan to reuse this information to further improve identification and quantification. We have added a new paragraph describing the limitations of the current method, see above.

11. How did the authors use DIA-NN solely for quantification? To my understanding, DIA-NN always performs an identification step first when starting from a spectral library. If the authors claim they only used DIA-NN for quantification, what additional parameters were used? Do they need to use the FDR cutoff calculated by DIA-NN? Have they considered developing their own quantification method rather than relying on DIA-NN?

Response: As DIA-NN is a library-based tool (the library is either from *in silico* prediction or from experimental data), we used it to perform targeted extraction and quantification in a library-based approach. Because we already have a high-quality spectral library, we need to disable the DIA-NN's match-between runs, which is designed for the *in silico* predicted library.

Do they need to use the FDR cutoff calculated by DIA-NN?

Response: We applied the FDR cutoff calculated by DIA-NN, as described in the Methods section. The overall FDR filtering is more stringent than the DIA-NN standalone version because there are 1% global PSM- and protein-level FDR cutoffs for spectral library generation and 1% global and run-wise precursor- and protein-level FDR cutoffs for DIA-NN quantification.

Have they considered developing their own quantification method rather than relying on DIA-NN?

Response: The diaTracer workflow in FragPipe can also be used with Skyline for quantification, which we mention in the text.

The spectral library is generated using EasyPQP, followed by the extraction of quantification from the DIA data using DIA-NN^{9, 18} (alternatively, using Skyline³⁵).

Reviewer #2 (Remarks on code availability):

The URL (<https://github.com/Nesvilab/diaTracer>) provides the README file with the instructions for installing and running test data. Instructions from the URL (<https://github.com/Nesvilab/diaTracer>) are not working in my laptop.

Response: Due to the limited information provided, we cannot identify the reasons. However, according to feedback from other reviews and many users, diaTracer worked for them. We encourage the users who have problems running our tools to submit a ticket to our GitHub repository.

Reviewer #3 (Remarks to the Author):

Reviewer #3 (Remarks on code availability):

The link <https://github.com/Nesvilab/diaTracer-manuscript> dose not work. Also, they didn't provide the source code of the algorithm.

Response: We have made the link <https://github.com/Nesvilab/diaTracer-manuscript> publicly available. There is source code to reproduce the figures. The code availability is described in the “code availability” section. FragPipe’s source code is available. diaTracer is not open source.

Reviewer #4 (Remarks to the Author):

I co-reviewed this manuscript with one of the reviewers who provided the listed reports. This is part of the Nature Communications initiative to facilitate training in peer review and to provide appropriate recognition for Early Career Researchers who co-review manuscripts

Reviewer #4 (Remarks on code availability):

Code is not available

Response: We have included the source code to reproduce the figures (see the “Code availability” section). FragPipe’s GUI source code is also available. The executable code is available for all tools that are a part of FragPipe. The source code for many of the tools is available as well. diaTracer is not open source.

Reviewer #5 (Remarks to the Author):

In this work, the authors present diaTracer for the spectrum-centric analysis of diaPASEF data. The diaTracer generates pseudo-MS/MS spectra based on XIC features between all detected precursors and fragments within user-defined ion mobility and retention time tolerance. Then the generated pseudo-MS/MS spectra were searched by FragPipe for further analysis, such as open search, PTMs analysis, etc. Examples are provided for the analysis of CSF data, plasma proteome, HLA immunopeptides, phosphopeptides, low-input and spatial proteomics.

Overall, the authors provide a nice tool in spectrum-centric analysis of DIA data. The main criticism is about the novelty of the work. The idea of diaTracer is an supplementary of DIA-Umpire, which was developed by the same group. The main difference is taking ion mobility into consideration by diaTracer. DIA-Umpire considers the elution profiles similarity to group precursors and fragments and then generates pseudo-MS/MS spectra. The diaTracer considers additionally the ion mobility information. After the generation of the pseudo-MS/MS spectra, all the analysis is the same as the standard FragPipe workflow. In view of this, the novelty of the work is doubtful for publication on Nature Communications, which normally publish work with significant novelty.

Response: We thank the reviewer for the review and summary. diaTracer has been designed to support the ion mobility data natively. The program was written from scratch and was not based on the DIA-Umpire code base. Although diaTracer builds on the concepts pioneered by DIA-Umpire (as do many other tools, including Spectronaut’s directDIA approach), there are innovations in diaTracer related to signal detection and extraction, as described in the manuscript.

In addition, there are some technique issues:

- The authors mentioned that Spectronaut supports direct identification from diaPASEF data. Although the algorithm is not disclosed, the tool should be compared with the newest version of Spectronaut 19 for benchmarking purpose. For instance, by analyzing the CSF data mentioned in this work, 1361 protein groups can be identified in average or 1600 in total using the directDIA mode of Spectronaut 19 (not the hybrid library mode, which can generate even more identification), more than the number of identifications discussed in this submission. Besides,

the version of Spectronaut which authors mentioned in HLA immunopeptidomics datasets is too old.

Response: We compared the results with those from Spectronaut when the results were made available as part of the original publication (e.g. Figure 4). For the other experiments, we were not able to perform the comparisons. Spectronaut is a commercial software that requires a paid license, even for academic scientists, to download. We do not have access to it. Furthermore, to the best of our knowledge, the Spectronaut's method for diaPASEF analysis has not been published in peer-reviewed literature (or anywhere in detail, really). Without the full description of their method and without knowing how the software works, we are not in a position to perform a fair comparison ourselves.

Our proposed method also supports the hybrid library mode, which results in the largest number of identifications, as described in the Results section and in Figure 2.

- Clinical samples are not suitable for benchmarking new tools. The authors should start from simple datasets like HeLa. The authors should further explain the purpose of selection of these datasets. Is there an internal logic to these selections?

Response: We have focused on datasets where there is a greatest need for a tool such as diaTracer. We used a variety of datasets, such as CSF, plasma, immunopeptidomics, and cell line phosphoproteomics data (with replicates and different gradient lengths) to benchmark the tools. These datasets require large search spaces (to account for nonspecific/semi-tryptic peptides or PTMs). We hope that the revised manuscript makes it clearer.

- The authors compared diaTracer with other works before. In most other works, they used DIA-NN originally. The authors need to check the version and parameters to ensure that they are consistent.

Response: We checked them; they are consistent.

- The accuracy should be compared in addition to the numbers of identification, for instance, the consistence between the spectrum-centric analysis and peptide-centric analysis of diaPASEF data, and the consistence with other software solutions, e.g. DIA-NN and Spectronaut. The accuracy and precision in quantification should also be compared.

Response: We thank the reviewer for the suggestion. We have generated the quantification correlation plots using four technical replicates of the phosphoproteome dataset (Figure 5b). We have also revised the text accordingly.

We also found that the quantification results reported by diaTracer-based direct diaPASEF workflow in FragPipe showed high correlations across four technical replicates (Figure 5b).

- In Methods, the authors should provide precise arithmetic formulas explaining the diaTracer algorithms to make the description more explicit. For example, the authors mentioned “data from adjacent neighbor scans are aggregated to form a composite 3D matrix, subsequently removing the data points lacking proximate neighbors.”. Firstly, the definition of distance should be provided in arithmetic formulas. Secondly, it doesn’t make sense when the authors mentioned “data points lacking proximate neighbors” as apparently all data points have proximate neighbors and what does matter in this scenario is the distance between data points and their proximate neighbors. Vague description like prementioned should be demonstrated more explicitly with arithmetic formulas.

Response: We thank the reviewer for the comment. We have revised the Methods and added more details.

The data from one isolation window ($1/k_0 \times m/z$) in each retention time (RT) frame is stored as a sparse matrix sized at $N \times W$, where W is the number of ion mobility (IM) bins determined by the IM window width and resolution. It equals the isolation window frame scan range divided by the smallest IM difference between two signals. N is the number of m/z bins, which is determined by the isolation window width and mass resolution. It equals the isolation window m/z range divided by the smallest m/z difference between two signals. To enhance the signal and reduce noise interference²⁵, data in the same isolation window of adjacent neighbor RT frames are aggregated to form a composite 3D matrix by summing the intensity of signals and subsequently removing the data points lacking proximate neighbors within a certain range. A 2D Gaussian filter is then employed to smooth the $N \times W$ matrix data. Local maxima are identified within the matrix and used as seeds for Gaussian fitting to determine 2D peaks. The center of the 2D Gaussian peak determines the m/z and IM values of the corresponding feature.

Minor opinions:

- Way too much bar plots are employed in the article, please use other ways to show the results.

Response: We have replaced Figure 4d with pie charts for the distribution of charge states and Figure 5b with correlation plots.

Reviewer #5 (Remarks on code availability):

The codes are clear.

RESPONSE TO REVIEWERS

Reviewer #5 (Remarks to the Author):

The authors have made some improvement by this revision. However, the main concerns are not fully addressed.

It is appreciated that the authors develop open-source or free software solutions for proteomics, which benefits the society. However, to publish an article on distinguished journals like Nature Communications, I believe extensive benchmarking analysis is required. The authors argued that Spectronaut is not free and doesn't publish their algorithm for diaPASEF analysis, which are true. Nevertheless, it is a software widely used for proteomics, including in the analysis of diaPASEF data. There is no evidence that the results from Spectronaut are not reliable. In view of this, I believe extensive comparison with Spectronaut is required to convince people using diaTracer. As the authors do not have access to Spectronaut software, I would suggest to perform more comparison with the published results using Spectronaut. A single comparison on one dataset can take bias. I have quickly searched through the diaPASEF datasets with Spectronaut directDIA analysis results public available, and found <https://doi.org/10.1038/s41597-024-03632-2> with Spectronaut 18.5 and <https://pubs.acs.org/doi/10.1021/acs.jproteome.2c00735> with Spectronaut 17.

Response: We thank the reviewer for these comments. We have added a new comparison recommended by the reviewer, with a very recent version of Spectronaut (version 18.5). We used the dataset suggested by the reviewer (<https://doi.org/10.1038/s41597-024-03632-2>). We added a new Figure 2 showing that FragPipe with diaTracer outperforms Spectronaut 18.5 in these data. The new results and methods are presented in the “Performance evaluation: Deep proteome profiling” and “Deep proteome profiling triple-negative breast cancer data analysis” sections, respectively.

We also note that our manuscript already contains comparisons between FragPipe and Spectronaut 17 in the phosphorylated peptide data and with Spectronaut 16 in the HLA data.

For the accuracy in quantification, the authors added the new analysis results in Figure 5b. However, there is a lack of comparison of the results with those from other analysis methods, including the spectral library based FragPipe DIA and DIA-NN, as well as Spectronaut. Coefficient of variation should be compared.

Response: We thank the reviewer for the comments. We have added the coefficient of variation comparison, which shows that FragPipe and Spectronaut perform similarly

(shown in the new Supplementary Figure 8).

*“We also compared the quantification performance with Spectronaut. **Supplementary Figure 8** shows the CV distribution, with box plots illustrating the quartiles and median of the CVs. The result reveal that our tool has comparable quantification precision with Spectronaut in these data.”*

Response to remarks on code availability

Reviewer #3 (Remarks on code availability):

Since they are not open-source, the code dose not contain any algorithm details.

Reviewer #5 (Remarks on code availability):

The authors only provided codes to reproduce the figures. diaTracer is not open source. The reviewer cannot further validate the software.

Response: As described in the manuscript, the code availability section, the executable codes are freely available for download and use under academic license. More specifically, the diaTracer executable code is available at <https://msfragger-upgrader.nesvilab.org/diatracer/>. It can also be downloaded from within the FragPipe GUI. As of October 14, 2024, the latest release of diaTracer (1.1.5) has been downloaded 3979 times. While diaTracer is not open source, the algorithm details are fully described in the manuscript.